# Enterococcal cell wall remodelling underpins pathogenesis via the release of the Enteroccocal Polysaccharide Antigen (EPA)

Robert E. Smith[1☯], Bartosz J. Michno[2,3☯], Rene L. Christena[1☯], Finn O'Dea[1], Jessica L. Davis[1], Ian D.E.A. Lidbury[1], Marcel G. Alamán-Zárate[1], Danai Stefanidi[1], Emmanuel Maes[4], Hannah Fisher[1], Tomasz K. Prajsnar[2], Stéphane Mesnage[1]*

**1** Molecular Microbiology, School of Biosciences, University of Sheffield, Sheffield, United Kingdom, **2** Department of Evolutionary Immunology, Institute of Zoology and Biomedical Research, Faculty of Biology, Jagiellonian University, Kraków, Poland, **3** Doctoral School of Exact and Natural Sciences, Jagiellonian University, Krakow, Poland, **4** University of Lille, CNRS, Inserm, CHU Lille, Institut Pasteur de Lille, Lille, France

☯ These authors contributed equally.
* s.mesnage@sheffield.ac.uk

## Abstract

Enterococci are opportunistic pathogens displaying a characteristic ovoid shape, typically forming pairs of cells (diplococci) and short chains. Control of cell chain length in *Enterococcus faecalis* relies on the activity of the major *N*-acetylglucosaminidase AtlA. The formation of short chains and diplococci is critical during pathogenesis for dissemination in the host and to limit recognition by innate immune effectors such as complement molecules and phagocytes. Here, we identify AtlE, an *N*-acetylmuramidase that contributes to septum cleavage during stationary phase in the absence of AtlA. AtlE is encoded by the locus required to produce the decoration subunits of the Enterococcal Polysaccharide Antigen (EPA), which mediate evasion of phagocytosis. We show that peptidoglycan hydrolysis by AtlE is essential for pathogenesis and demonstrate that soluble cell wall fragments containing EPA decorations increase the virulence of *E. faecalis*, suggesting that EPA plays a role as a decoy molecule to evade host defences. This research sheds light on the complex interplay between bacterial cell division, cell wall remodelling, and the host immune system, providing valuable insights into a novel mechanism underlying the virulence of *E. faecalis*.

## Author summary

The major component of the bacterial cell envelope (peptidoglycan) undergoes partial hydrolysis during growth. This process, referred to as remodelling, is required for the incorporation of novel peptidoglycan building blocks, and cell separation at the end of division. In *Enterococcus faecalis,* only one ubiquitous peptidoglycan hydrolase, named AtlA, has been described so far. AtlA plays a

which permits unrestricted use, distribution, and reproduction in any medium, provided the original author and source are credited.

**Data availability statement:** LC-MS/MS data related to Figs 5 and S4 is available through the Glycopost repository (GPST000505; http://doi.org/10.50821/GLYCOPOST-GPST000505). All data used in this work are provided as supplementary information.

**Funding:** RLC was the recipient of a Commonwealth Rutherford fellowship (INRF-2017-163). MGAZ was supported by a Conacyt studentship from the Mexican government (2021-000007-01EXTF-00221). JLD, RES and HF were funded by the White Rose Doctoral Training Programme (BBSRC grant BB/M011151/1); FOD was a recipient of a PhD studentship White Rose Doctoral Training Programme (BBSRC grant BB/T007222/1). BJM and TKP were supported by National Science Centre of Poland within Sonata Bis 9 project (Grant number UMO-2019/34/E/NZ6/00137). The funders had no role in study design, data collection and analysis, decision to publish, or preparation of the manuscript.

**Competing interests:** The authors have no competing interests to declare.

prominent role in septum cleavage and is responsible for the characteristic formation of diplococci and short cell chains. The minimization of cell chain length by AtlA is critical for innate immune evasion and underpins pathogenesis. Here, we identify another ubiquitous peptidoglycan hydrolase named AtlE encoded by the Enterococcal Polysaccharide Antigen (EPA) biosynthetic locus. We show that AtlE displays *N*-acetylmuramidase activity and requires strain-specific EPA decorations to be active. Whilst AtlE only plays a marginal role in septum cleavage during growth, AtlE is essential for virulence in the zebrafish model of infection. We demonstrate that AtlE activity contributes to release cell wall fragments and promotes phagocyte evasion, indicating that EPA plays a role as a decoy molecule produced by enterococci to counteract host immune defenses.

## Introduction

Peptidoglycan is a bag-shaped macromolecule that represents the major and essential component of the bacterial cell envelope [1,2]. It confers cell shape and resistance to internal osmotic pressure. In monoderm bacteria, which lack an outer membrane, peptidoglycan serves as a scaffold for the display of proteins and surface polymers [3]. Peptidoglycan is made of glycan chains that consist of *N*-acetylglucosamine and *N*-acetylmuramic acid, crosslinked by short peptide stems [1]. The assembly of this macromolecule is a complex process involving dynamic protein complexes interacting with cytoskeletal elements, achieving both cell elongation and division [4]. During growth and division, peptidoglycan undergoes limited hydrolysis to allow the insertion of novel building blocks and the cleavage of the septum to separate and release daughter cells. The spatio-temporal control of peptidoglycan remodelling by hydrolases is crucial to avoid cell lysis and maintain cell morphology. It involves several co-existing mechanisms, including transcriptional and post-translational control, such as proteolysis or activation via protein-protein interactions (for a recent review, see [5]).

In *Enterococcus faecalis,* a peptidoglycan hydrolase called AtlA plays a prominent role in septum cleavage [6]. This enzyme can cause rapid autolysis following carbon source depletion [7] and its activity is therefore tightly controlled via a combination of mechanisms that underpin subcellular targeting and enzymatic activation. AtlA is specifically recruited within the cytoplasm to the septum via its C-terminal domain [8]. This recruitment is facilitated by an unusually long signal peptide (53 residues) similar to extended signal peptide regions (ESPR) [9,10] shown to slow down translocation across the cytoplasmic membrane [8]. Once exposed at the cell surface, AtlA undergoes proteolytic cleavage by extracellular proteases. This step removes the glycosylated *N*-terminal domain, likely limiting access to the substrate through steric hindrance [11]. Finally, six C-terminal repeats have been proposed to bind preferentially and cooperatively to denuded glycan chains at the septum, ensuring efficient and restricted peptidoglycan hydrolysis [12,13].

The exquisite regulation of AtlA activity is critical to maintain the characteristic morphology of *E. faecalis*, which forms mostly diplococci and short chains containing 4–8 cells [14]. In the absence of AtlA activity, cells form longer chains which are readily recognised by phagocytes, and the virulence is abolished [11], indicating that minimizing cell chain length is an important process for enterococci.

Two other peptidoglycan hydrolases, AtlB and AtlC have also been identified and shown to play a minor role in septum cleavage. A striking observation was that in the absence of AtlA, AtlB and AtlC, the extremely long cell chains formed in exponential phase reverted to form diplococci or short chains during late stationary phase [6]. The enzyme(s) responsible for this phenomenon remained unknown.

Here, we identify a peptidoglycan hydrolase (AtlE), that contributes to daughter cell separation in late stationary phase in the absence of AtlA. AtlE is encoded within the Enterococcal Polysaccharide Antigen (EPA) locus, flanked by the genes responsible for the biosynthesis of EPA decoration subunits, essential for pathogenesis [15]. We reveal that distinctive abundance and activity of AtlA and AtlE during growth contribute to their specific roles during exponential and stationary phase. Whilst the deletion of *atlE* has no impact on EPA structure or its cell surface exposition (as shown by HR-MAS NMR), we show that this mutation abolishes virulence in the zebrafish model of infection, suggesting that the digestion of enterococcal cell walls is important for pathogenesis. In agreement with this hypothesis, we show that soluble cell wall fragments containing EPA decorations play a role as decoy molecules to cause infections and death.

## Results

### A novel peptidoglycan hydrolytic activity is detected in culture supernatants of the *E. faecalis* Δ*atlABC* mutant

The chains formed by the *E. faecalis* JH2–2 Δ*atlABC* triple mutant in exponential phase [6] revert to diplococci and short chains in stationary phase, indicating that other enzyme(s) cleave the septum during this growth phase. No enzyme candidate for this activity could be identified by zymogram in crude extracts and supernatants from the Δ*atlABC* mutant using the parental cell walls as a substrate. We therefore repeated zymogram experiments using a less crosslinked and more susceptible substrate corresponding to autoclaved cells from a mutant harbouring multiple deletions in class A PBP genes [16] (Fig 1A). After 72h of incubation in renaturation buffer, a band with peptidoglycan hydrolytic activity was detected in Δ*atlABC* mutant.

To identify the corresponding autolysin, stationary phase culture supernatants were TCA-precipitated, and protein bands detected on a Coomassie-stained SDS-PAGE were individually excised and analysed by LC-MS/MS following in-gel tryptic digestion (Fig 1B and Table 1). A search against the annotated proteome of the reference *E. faecalis* strain V583 identified three candidates in the band matching the size of the hydrolytic activity on the zymogram: EF0252 and EF0114 (two putative *N*-acetylglucosaminidases) and EF2174 (a putative *N*-acetylmuramidase).

### Identification of the gene encoding AtlE

Genes encoding the three putative peptidoglycan hydrolases identified in Δ*atlABC* mutant supernatants (Fig 1B) were inactivated by allelic exchange in the Δ*atlABC* background. Whilst no change was observed in the zymogram profiles following the deletion of genes homologous to V583 *EF0252* and *EF0114* (S1 Fig), no band was detected after the deletion of the *EF2174* homolog (Fig 2). As expected, complementation of the *atlE* deletion restored the production of a hydrolytic band in the supernatants of the quadruple mutant, and the activity in the complemented strain increased in the presence of a higher concentration of anhydrotetracycline inducer (Fig 2, lanes 4–6). Hereafter, we name this gene *atlE*.

### AtlE contributes to septum cleavage in the absence of AtlA but plays a marginal role in cell separation

We sought to further investigate the activity of AtlE produced by the pathogenic *E. faecalis* strain OG1RF, since the composition of this bacterium's cell envelope has been more thoroughly characterised than JH2–2. Furthermore, OG1RF is a clinical isolate used to explore pathogenesis in various models of infection [17–19]. OG1RF lacks *atlB* and *atlC* but

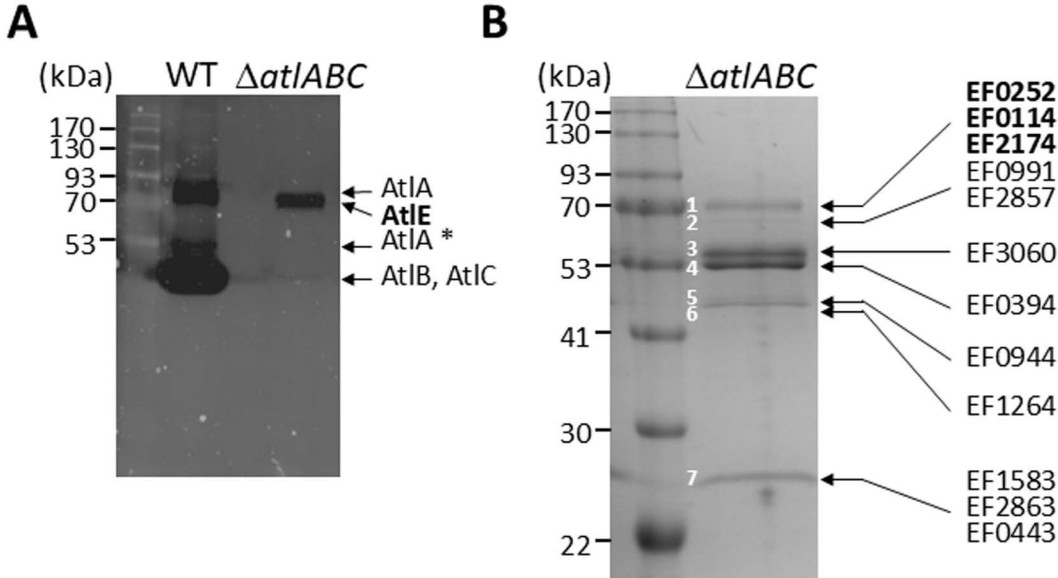

**Fig 1. Identification of a novel peptidoglycan hydrolase activity and secreted proteins in *E. faecalis* JH2-2 culture supernatants. A**, Detection of peptidoglycan hydrolytic activities in culture supernatants. 20 µl of culture supernatants from the JH2-2 (WT) and Δ*atlABC* isogenic mutant grown over-night were loaded on an SDS-PAGE containing autoclaved cells of a mutant with reduced peptidoglycan cross-linking (triple class A PBP mutant Δ*ponA* Δ*pbpF* Δ*pbpZ*). Zymograms were incubated for 72h at 37°C The enzyme with hydrolytic activity present in the triple Δ*atlABC* was named AtlE; the band corresponding to AtlA with a truncated N-terminal domain is indicated with an asterisk. **B**, SDS-PAGE analysis of TCA-precipitated supernatants from the Δ*atlABC* mutant. The proteins corresponding to the seven bands detected were cut and digested with trypsin for LC-MS/MS analysis. The sequence of peptides resulting from trypsin digestions were used to identify *E. faecalis* V583 homologs. Candidates with potential peptidoglycan hydrolytic activity (EF0252, EF0114, EF2174) are indicated in bold.

contains *atlA* and *atlE*. In both strains, *atlE* is part of the locus encoding the Enterococcal Polysaccharide Antigen (EPA), polymer covalently anchored to peptidoglycan (S2 Fig).

Zymogram analysis of *altA* and *atlE* deletion mutants confirmed the absence of the specific autolytic bands corresponding to their activities and the double mutant Δ*atlA* Δ*atlE* did not contain any lytic band. (S3 Fig, lanes 1–4). Complementation of Δ*atlA* Δ*atlE* with a plasmid-encoded copy of the *atlE* gene restored the production of the AtlE lytic band (S3 Fig, lane 5).

The role of AtlE in septum cleavage was then investigated using flow cytometry, measuring forward scattered light as a proxy for bacterial cell chain length during both the exponential and stationary phases of growth (Fig 3) [11]. In exponentially growing cells, *atlA* deletion led to a significant increase in cell chain length ($P<0.001$) but the *atlE* deletion alone or in combination with Δ*atlA* had no impact (Fig 3A, grey bars). As previously reported, the Δ*atlA* deletion was associated with a less pronounced phenotype in stationary phase, confirming that this autolysin is mostly active in exponential phase (Fig 3A, pink bars). The deletion of *atlE* alone had no impact on chain length but led to very clear increase in cell chain length when combined with the Δ*atlA* deletion ($P<0.001$). The chain forming phenotype was complemented in the Δ*atlA* Δ*atlE* mutant and increasing *atlE* expression led to a progressive reduction of cell chain chaining (Fig 3B). Collectively, these data confirmed that AtlE is a peptidoglycan hydrolase which plays a marginal role in septum cleavage to maintain the characteristic short chains and diplococci. The activity of this enzyme is more pronounced during stationary phase.

### *atlE* is a conserved gene in the EPA decoration locus but shows allelic variation in its C-terminal cell wall binding domain

The *atlE* gene is part of the locus encoding EPA decorations (*epa_var*) that varies across *Enterococcus* strains, resulting in the production of strain-specific EPA decoration structures [20,21]. A comparison of the *epa_var* region across *E.*

Table 1. LC-MS/MS identification of proteins secreted in JH2-2 Δat/ABC supernatants.

| Band | ID | Unique Peptides | iBAQ a | Intensity | Coverage | Size (kDa, aa) b | Role | Reference |
|---|---|---|---|---|---|---|---|---|
| 1 | EF0252 | 33 | 6.5 E+08 | 1.0 E+10 | 78.4 | 51.9, 477 | Peptidoglycan hydrolase (AtlD; GH73) | https://doi.org/10.1159/000486757 |
| | EF2174 | 18 | 3.1 E+08 | 4.7 E+09 | 46.3 | 73.1, 676 | Putative peptidoglycan hydrolase (GH25) | N/A c |
| | EF0114 | 41 | 4.4 E+07 | 1.7 E+09 | 47.2 | 94.0, 835 | endo-β-N-acetylglucosaminidases (GH18) | https://doi.org/10.1128/JB.00371-21 |
| 2 | EF0991 | 70 | 1.1 E+09 | 4.7 E+10 | 84 | 81.7, 742 | Class B Penicillin-binding protein (PBPc) | https://doi.org/10.1111/j.1574-6976.2007.00098.x |
| | EF0252 | 26 | 1.0 E+09 | 1.6 E+10 | 75.7 | 51.9, 477 | Peptidoglycan hydrolase (AtlD; GH73) | https://doi.org/10.1159/000486757 |
| | EF2857 | 55 | 4.7 E+08 | 1.7 E+10 | 59 | 711; 77.6 | Class B Penicillin-binding protein (PBP2B) | https://doi.org/10.1111/j.1574-6976.2007.00098.x |
| 3 | EF0360 | 53 | 1.1 E+09 | 4.9 E+10 | 71.2 | 48.3, 455 | Peptidoglycan hydrolase (SagA; Nlp/p60) | https://doi.org/10.7554/eLife.45343 |
| | EF0394 | 34 | 1.0 E+09 | 2.4 E+10 | 57.4 | 45.3, 428 | Secreted stress-response antigen (SalB) | https://doi.org/10.1128/JB.188.7.2636-2645.2006 |
| | EF0907 | 29 | 4.7 E+08 | 2.6 E+08 | 61.9 | 58.5, 534 | Hypothetical peptide ABC transporter | N/A |
| 4 | EF0394 | 35 | 3.4 E+09 | 4.7 E+10 | 52.5 | 45.3, 428 | Secreted stress-response antigen (SalB) | https://doi.org/10.1128/JB.188.7.2636-2645.2006 |
| | EF3060 | 21 | 1.2 E+08 | 2.4 E+09 | 55.2 | 48.3, 455 | Peptidoglycan hydrolase (SagA; Nlp/p60) | https://doi.org/10.7554/eLife.45343 |
| | EF0991 | 18 | 3.1 E+06 | 1.3 E+08 | 37.7 | 81.7, 742 | Class B Penicillin-binding protein (PBPc) | https://doi.org/10.1111/j.1574-6976.2007.00098.x |
| 5 | EF0944 | 44 | 3.3 E+09 | 5.6 E+10 | 60.3 | 40.7, 377 | Hypothetical secreted protein | N/A |
| | EF2860 | 39 | 6.0 E+08 | 1.4 E+10 | 71.5 | 53.6, 481 | L,D-transpeptidase (Ldt$_{fs}$) | doi.org/10.1074/jbc.M610911200 |
| | EF1264 | 32 | 3.9 E+08 | 1.5 E+10 | 49.1 | 80.0, 702 | Hypothetical sulfatase | N/A |
| 6 | EF1264 | 39 | 1.1 E+09 | 4.3 E+10 | 51.7 | 80.0, 702 | Hypothetical sulfatase | N/A |
| | EF0944 | 40 | 5.7 E+08 | 9.7 E+09 | 58.9 | 40.7, 377 | Hypothetical secreted protein | N/A |
| | EF0394 | 27 | 4.5 E+08 | 6.3 E+09 | 56.5 | 45.3, 428 | Secreted stress-response antigen (SalB) | https://doi.org/10.1128/JB.188.7.2636-2645.2006 |
| 7 | EF1583 | 6 | 2.6 E+08 | 2.1 E+09 | 30,1 | 27.1, 257 | Putative peptidoglycan hydrolase (GH73) | N/A |
| | EF2863 | 11 | 2.2 E+08 | 2.2 E+09 | 39,8 | 30.1, 273 | endo-β-N-acetylglucosaminidases (GH18) | https://doi.org/10.1111/j.1574-6968.2011.02419.x |
| | EF0443 | 4 | 1.7 E+08 | 1.4 E+09 | 27,1 | 18.7, 172 | LysM secreted protein | N/A |

a sum of all the peptides intensities divided by the number of observable peptides of a protein.

b Size of the mature, secreted protein as predicted by SignalP 5.0 (https://services.healthtech.dtu.dk/service.php?SignalP-5.0).

c N/A, not applicable.

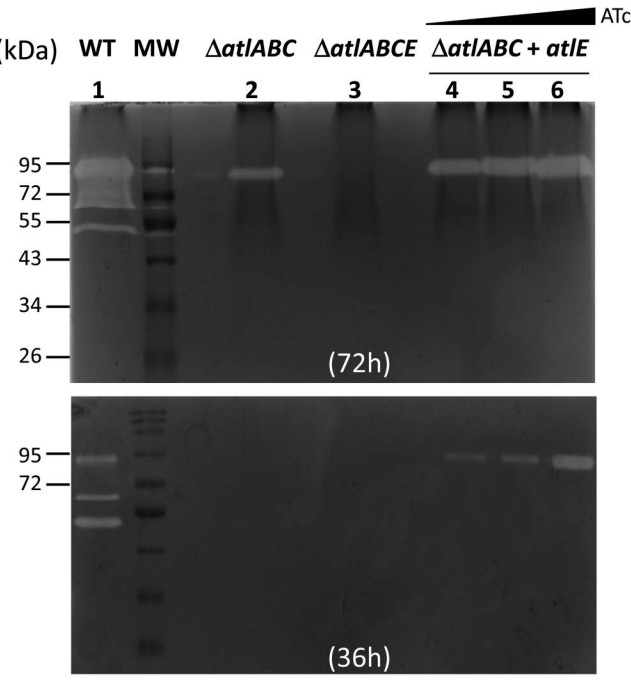

**Fig 2. Zymogram analysis of *E. faecalis* culture supernatants.** Peptidoglycan hydrolytic activities were detected in 20 µL of culture supernatants from strains JH2-2 (WT, lane 1), ΔatlABC (lane 2), ΔatlABCE (lane 3) and ΔatlABC + atlE (complemented ΔatlABC mutant, lanes 4-6) grown overnight. Cells from the triple class A PBP mutant ΔponA ΔpbpF ΔpbpZ were used as a substrate. The expression of *atlE* was detected in the absence (lane 4) or presence of anhydrotetracycline (ATc) at a concentration of 10 ng/µL (lane 5) or 200 ng/µL (lane 6). The zymogram was repeated with a renaturation time of 36h instead of 72h to see the increased amount of activity associated with *atlE* induction.

faecalis strains revealed that *atlE* is systematically present in loci with no sequencing gap (Fig 4A). AtlE is a multidomain protein that always contains an N-terminal glycosyl hydrolase domain belonging to the GH25 family (CAZy database; [22]). GH25 domains are found in enzymes displaying *N*-acetylmuramidase activity like the commercially available muta-nolysin (cellosyl). In contrast, the C-terminus is heterogeneous containing a variable number of imperfect repeats corresponding to either GW (pfam13457) or DUF5776 (pfam19087) domains (Fig 4A and 4B). GW and DUF5776 domains are structurally related to SH3 domains, which are involved in binding to cell envelope polymers. GW repeats have been associated with binding to LTAs [23], but no information is available about DUF5776.

Phylogenetic analyses revealed that the N-terminus containing the GH25 domain is partitioned into four subclades (Fig 4A). Clades I and II possess the GW C-terminal domain and clades III and IV possess the DUF5776 C-terminal domain, suggesting divergence in overall functionality and/or mechanism. Furthermore, the number of C-terminus repeating GW and DUF5776 domains differs between strains (Fig 4B), as does overall C-terminus protein identity (Fig 4A, outer ring). For example, OG1RF and V583 C-terminal domains are only 47% identical to each other, despite both enzymes belonging to clade IV and possessing the DUF5776 domain. Both GW and DUF5776 domains are made of a variable number of imperfect repeats, some of which displaying low sequence identity. For example, the first repeat of the GW domain of the OG1RF AtlE allele is 47% identical to the last repeat; repeats 1 and 2 of the JH2–2 allele are only 23% identical. In both cases, the Alphafold predicted structures for repeats present in the same GW or DUF5776 domain remain remarkably conserved (Fig 4C). By contrast, the predicted structures of GW and DUF5776 repeats revealed that these domains share distinct folds (Fig 4C).

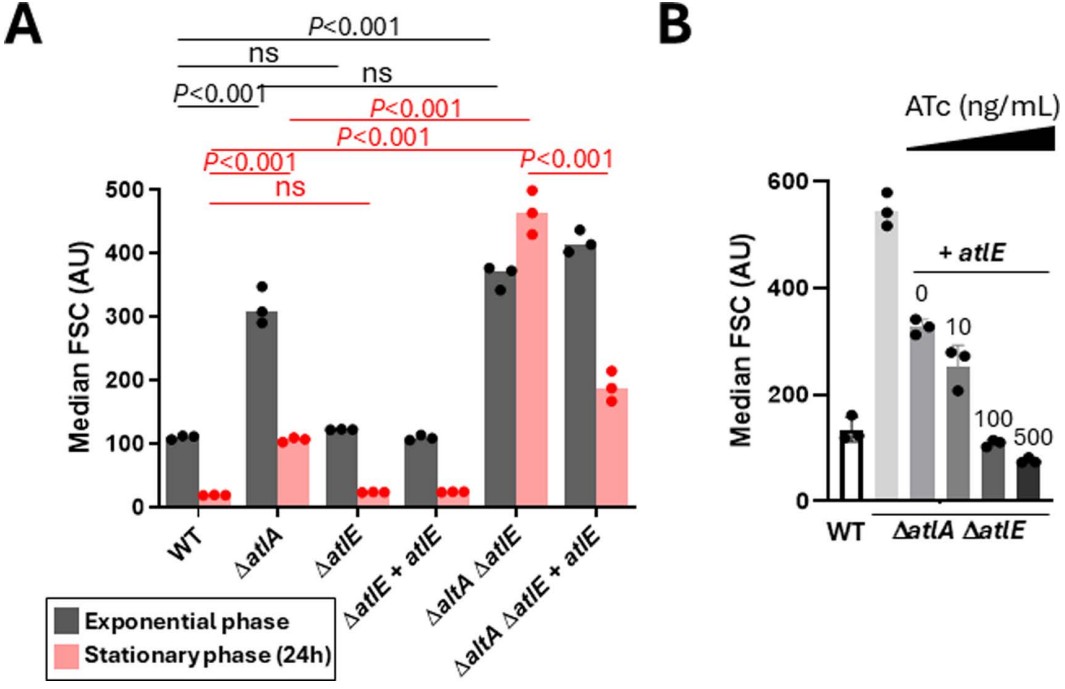

**Fig 3. Contribution of AtlE to septum cleavage and bacterial cell chain length. A**, Comparison of median forward scattered (FSC) light values corresponding to the cell chain lengths of OG1RF (WT), ΔatlA, ΔatlE, ΔatlE + atlE (complemented deletion strain), ΔatlA ΔatlE and ΔatlA ΔatlE + atlE strains in exponentially growing cells and after 24h of growth. For complementation experiments, precultures were grown in 10 ng/mL ATc then 50 ng/mL over the course of the experiment. Statistical analyses were carried out on results from biological triplicates using one-way ANOVA (Tukey's multiple comparison test). For exponential cells (grey bars): WT versus ΔatlA, P < 0.001; WT versus ΔatlE, ns, P = 0.93; WT versus ΔatlA ΔatlE, P < 0.001; ΔatlA versus ΔatlA ΔatlE, ns, 0.28. For stationary cells (24h cultures, pink bars): WT versus ΔatlA, P < 0.001; WT versus ΔatlE, ns, P > 0.99; WT versus ΔatlA ΔatlE, P < 0.001; ΔatlA versus ΔatlA ΔatlE, P < 0.001; ΔatlA ΔatlE versus ΔatlA ΔatlE + atlE, P < 0.001.. **B**, AtlE-mediated cleavage of bacterial septum. Median FSC of WT, ΔatlA ΔatlE mutant and complemented ΔatlA ΔatlE + atlE derivative cells was measured after 16h of growth (overnight cultures). Anhydrotetracycline (ATc) was added at various concentrations (0, 10, 200 and 500 ng/mL) to induce the expression of atlE.

### AtlE is a muramidase that cleaves peptidoglycan in a strain specific manner

We first sought to characterize the enzymatic activity of OG1RF AtlE (AtlE_O). AtlE_O, devoid of its signal peptide (residues 1–24) was produced recombinantly in *E. coli*, purified, and used to digest OG1RF cell walls (Fig 5A, insert). The resulting digestion products were analysed by LC-MS and LC-MS/MS. The muropeptide profile observed was similar to the profile resulting from digestion with mutanolysin [24], and major peaks were assigned to peptidoglycan fragments resulting from glycosyl hydrolase activity (Fig 5B and S4 Fig). In-source decay (Fig 5C) and LC-MS/MS analysis (S4 Fig) confirmed that AtlE displays *N*-acetylmuramidase activity, releasing Mur*N*Ac residues at the reducing end of disaccharide-peptides.

Unlike the catalytic GH25 domain ubiquitously found at the N-terminus of AtlE, the *C*-terminal cell wall binding domain of this enzyme varies across isolates (Fig 4A), suggesting that it could specifically recognise strain-specific cell wall components such as EPA decorations. In agreement with this hypothesis, AtlE_O was able to digest cognate OG1RF cell walls, giving a similar profile to mutanolysin (Fig 6A), but displayed very poor activity (if any) against JH2–2 cell walls (Fig 6B). To test the contribution of EPA to AtlE activity, we used cell walls from *E. faecalis* Δ11720 as a substrate [20]. Δ11720 cells produce EPA which contains a rhamnose backbone only substituted by two residues (*N*-acetylgalactosamine and glucose) and therefore does not contain any repeating unit making EPA decorations [20]. No activity was detected against

PLOS Pathogens

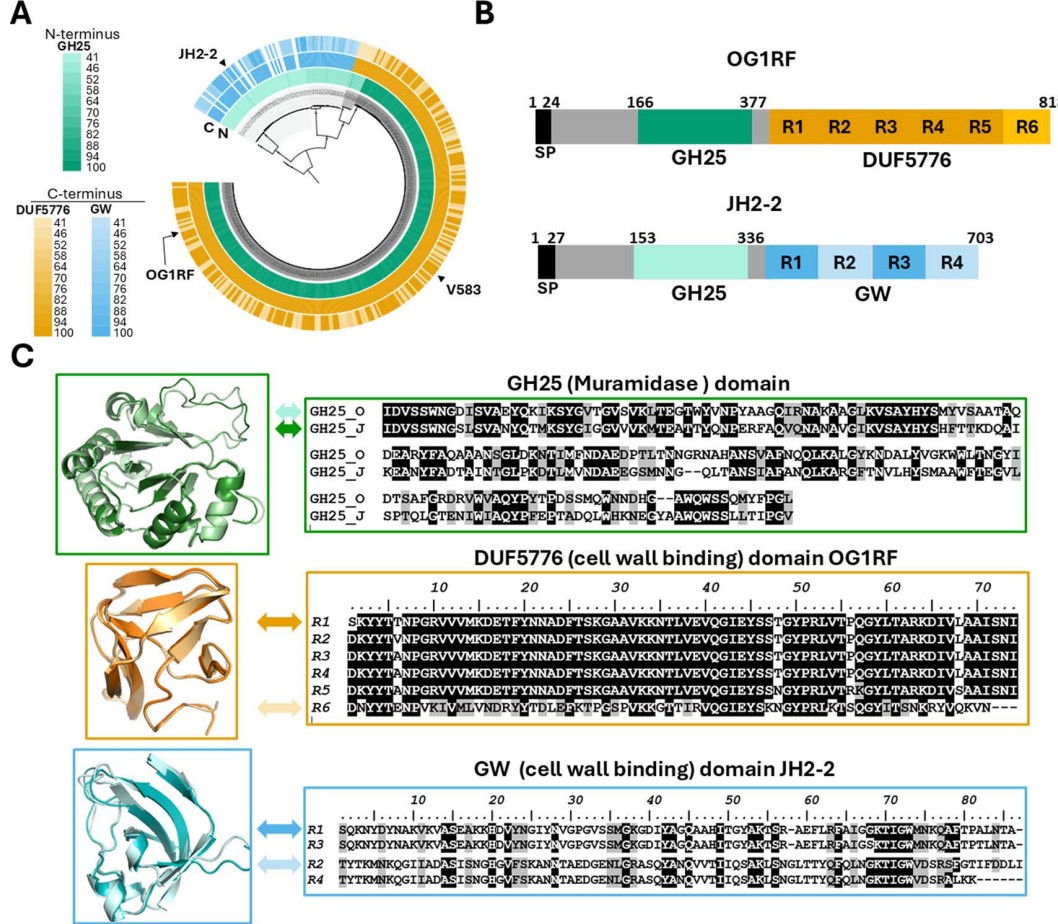

**Fig 4. Distribution and allelic variation of the *E. faecalis atlE* gene. A**, Phylogenetic reconstruction of AtlE identified in *E. faecalis* genomes. Open reading frames were trimmed to contain only the GH25 domain, totalling 196 sites of which 167 were parsimony informative. Evolutionary distance was inferred using the Maximum Likelihood method using the WAG+G4 best fit model. *E. faecalis* V583 GH25 (N-terminus, inner ring) and DUF5776 (C-terminus, outer ring, green) domains and *E. faecalis* JH2-2 GW (C-terminus, outer ring, blue) domains were used as references for sequence comparisons. **B**, Domain organisation of OG1RF and JH2-2 AtlE alleles. Residues numbers are indicated on the top of the sequence; Individual DUF5776 and GW repeats are numbered from N to C-terminus. SP, signal peptide. **C**, Alphafold structure predictions of GH25, DUF5776 and GW domains from OG1RF and JH2-2. The predicted structures of two DUF5776 and GW domains showing the less similar sequences were aligned (R1 and R6 for OG1RF, R1 and R2 for JH2-2).

cell walls extracted from the *Δ11720* mutant (Fig 6C). Collectively, these data indicate that the presence of EPA decorations are required for the activity of AtlE.

## AtlA and AtlE expression levels and differential activities against cell walls reflect distinct roles during growth

We next investigated why AtlA activity is predominant during exponential growth, whilst AtlE activity can only be detected in stationary phase (Fig 3). The level of expression of both proteins during exponential and stationary phase was compared using specific antibodies (S5 Fig). Cell-associated proteins from cultures in exponential and stationary phase (OD ≈ 0.3 and OD ≈ 2, respectively) were serially diluted and probed with anti-AtlA and anti-AtlE antibodies (Fig 7A), showing that AtlA abundance was strikingly higher during exponential phase. By contrast, AtlE was present in similar amounts during exponential and stationary phase. We next measured the peptidoglycan hydrolytic activity of AtlA and AtlE enzymes

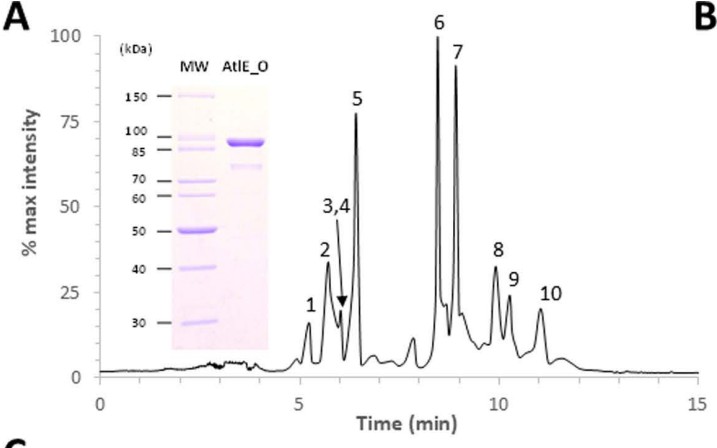

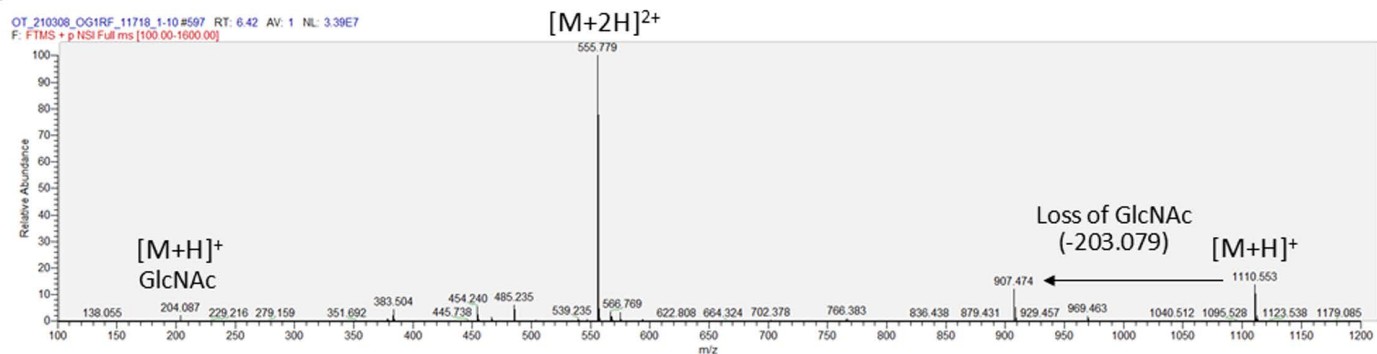

**Fig 5. LC-MS characterisation of AtlE enzymatic activity. A**, Total Ion Count chromatogram corresponding to the OG1RF muropeptide solubilised by AtlE_O. Major peaks are numbered. The inset shows an SDS-PAGE of AtlE_O purified by immobilised metal-affinity chromatography. **B**, Identification of major muropeptides shown in **A. C**, Extracted ion chromatogram corresponding to the major monomer (peak 5), showing singly and doubly charged ions matching the expected *m/z* value for a disaccharide heptapeptide containing a pentapeptide stem and two L-Alanine residues in the lateral chain. The loss of a non-reduced GlcNAc residue shows the *N*-acetylmuramidase activity of the enzyme.

*in vitro* against purified cell walls. Both enzymes were more active on cell walls from exponentially growing cells, but AtlA had a much more reduced activity on cell walls from stationary growing cells when compared to AtlE (Fig 7B). Treatment of cell walls with HCl, which removes polysaccharides covalently bound to PG, abolished AtlE activity (Fig 7C). AtlA was less active on pure PG extracted from exponential phase but more active on PG extracted from stationary phase (Fig 7C). Collectively, these experiments indicated that the predominant role of AtlA during exponential growth is underpinned by a higher protein abundance and a preferential activity on cell walls from exponentially growing cells.

## AtlE does not contribute to the surface exposure or structure of EPA

EPA decorations are surface exposed and form a pellicle around the cell [25]. We hypothesised that AtlE activity may play a role in the surface exposure of EPA decorations, trimming away the surrounding peptidoglycan. To test this hypothesis, we used high-resolution magic angle spinning (HR-MAS) NMR (S6A Fig). This technique is performed on whole cells and specifically allows the detection of molecules that are in highly flexible, solvent-exposed environments; molecules with less rotational freedom and/or are not solvent-accessible are not detected. $^1$H-$^{13}$C HSQC HR-MAS NMR experiments recorded on *E. faecalis* OG1RF and Δ*atlE* cells showed that AtlE does not contribute to the production or display of surface-exposed EPA or lipoteichoic acid (LTA) [26]. Two other unidentified cell wall polysaccharides previously reported

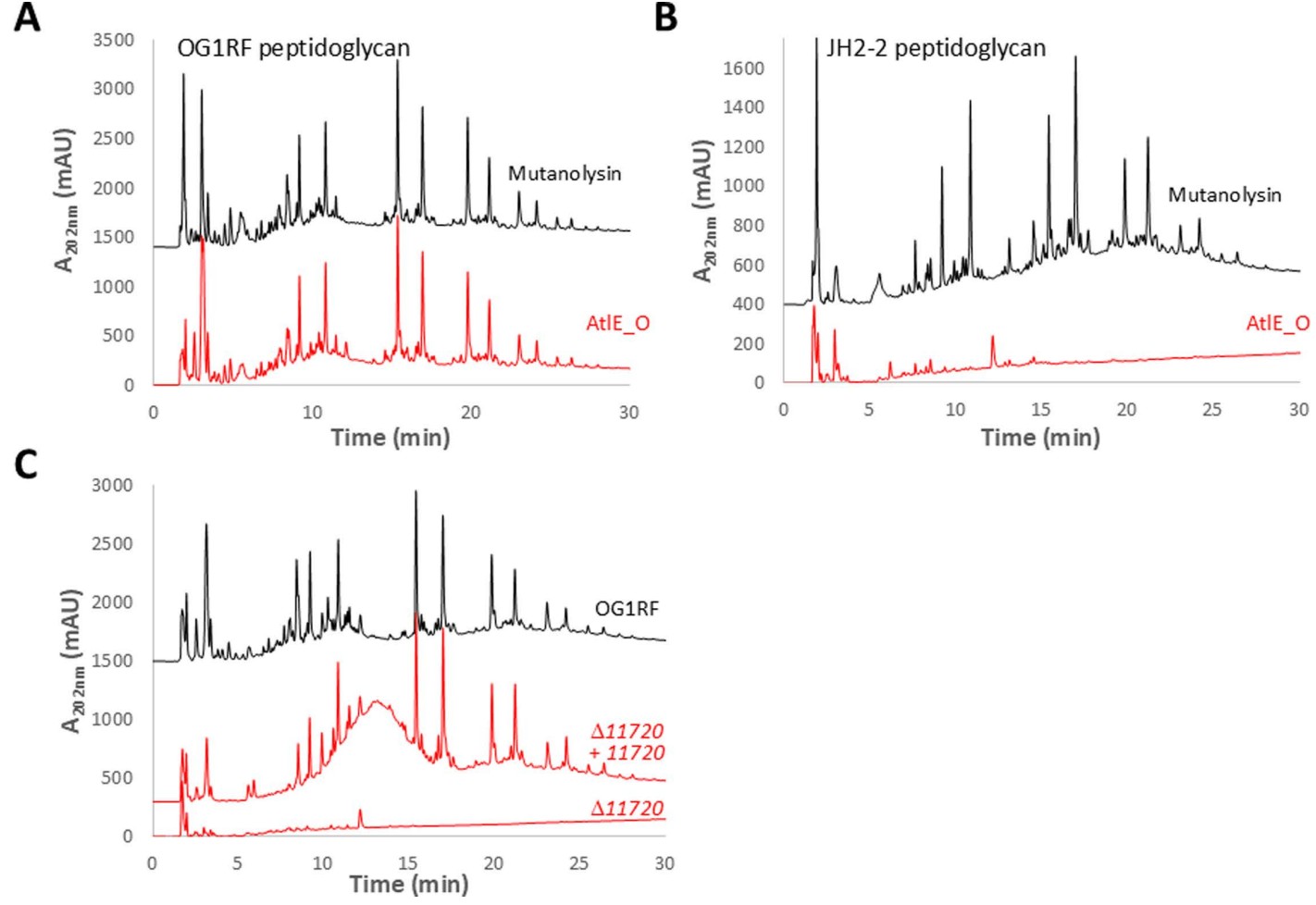

**Fig 6. HPLC analysis of AtlE_O substrate specificity.** The activity of recombinant AtlE_O was tested against several *E. faecalis* cell walls. Muropeptide profiles resulting from the digestion of OG1RF (**A**) or JH2-2 cell walls (**B**) with mutanolysin (black) and AtlE_O (red). **C,** Digestion of cell walls from OG1RF (black) and EPA decoration mutant *Δ11720* and complemented derivative (red).

[21] (denoted with an asterisk in S6A Fig) were also detected, and the corresponding signals remained unchanged in the Δ*atlE* mutant. 2D NMR experiments on purified EPA confirmed that the structure of the polysaccharide produced by the mutant was identical to the parental OG1RF (S6B Fig). Finally, electron micrographs of thin sections revealed the presence of a pellicle at the cell surface of both OG1RF and the Δ*atlE* mutant (S6C Fig), indicating that no major changes in the cell surface architecture are associated with the deletion of *atlE*.

### AtlE contributes to the release of EPA in culture supernatants during cell wall remodeling

Previous studies revealed that peptidoglycan fragments are released into culture supernatants during growth [6]. To test the contribution of AtlE to this process, we grew the parental OG1RF strain and its Δ*atlE* derivative in chemically defined medium. Culture supernatants harvested in stationary phase were filtered, concentrated and analysed by NMR to detect the presence of EPA (Fig 8). Phosphorus NMR revealed the presence of signals between -2–2 ppm expected for EPA absent in the samples prepared from the mutant (Fig 8A). Methyl rhamnose signals with a resonance between 1.2-1.5ppm were also detected in OG1RF but not in the mutant (Fig 8B). In agreement with these results, the comparison of 1D $^{1}$H

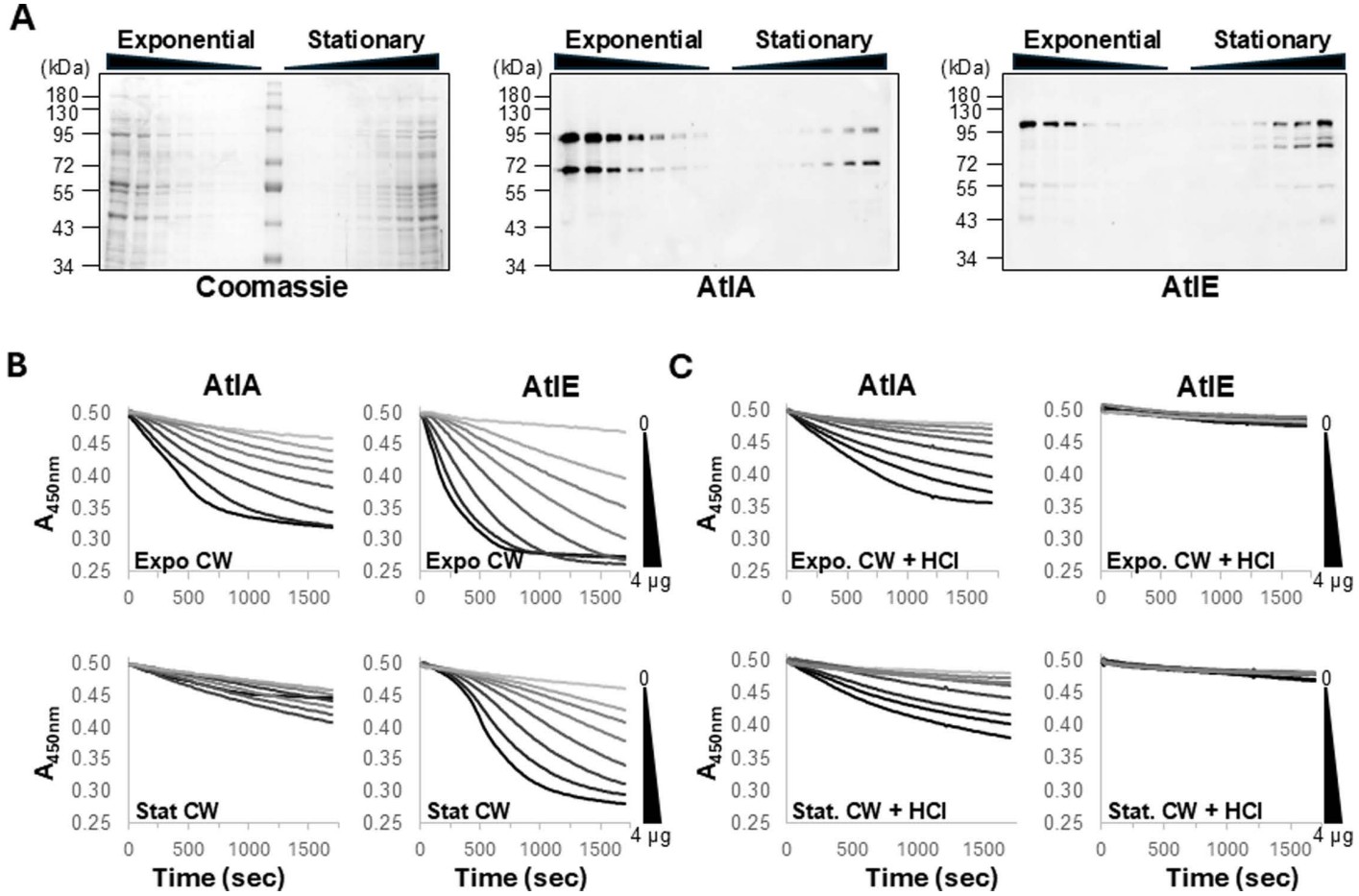

**Fig 7. Abundance and activity of AtlA and AtlE during growth. A**, Crude extracts prepared from cells in exponential ($OD_{600nm}$≈0.3) and stationary ($OD_{600nm}$≈2.0) cultures were serially diluted and probed with specific antibodies raised against AtlA and AtlE. Peptidoglycan hydrolytic activities of AtlA and AtlE were measured against purified cell walls from exponential ($OD_{600nm}$≈0.3) and stationary cultures ($OD_{600nm}$≈2.0) before (**B**) and after (**C**) treatment with 1 M HCl to remove cell wall polymers. Each assay contained 0, 0.625, 0.125, 0.25, 0.5, 1, 2, and 4 µg of recombinant protein and cell walls adjusted to an Absorbance at 450nm of *c.a.* 0.5. Initial Absorbance values were adjusted to 0.5 for comparison purposes. Cell wall hydrolysis was measured by following the decrease in absorbance over a period of 20 min.

and 2D $^1$H-$^{13}$C HSQC spectra of OG1RF and Δ*atlE* samples revealed anomeric signals for OG1RF but virtually none for the mutant (Fig 8C–8D). Collectively, this analysis therefore indicated that AtlE activity contributes to the release of EPA in culture supernatants.

### Exploring the role of AtlE activity during pathogenesis in the zebrafish model of infection

Both the minimization of cell chain length and the production of EPA mediate innate immune evasion and are critical for infection [11,27]. A preliminary experiment revealed that the lack of AtlE activity has a significant effect on virulence (S7 Fig). Depletion of phagocytes using *pu.1* morpholino restored the virulence of the Δ*atlE* mutant to parental levels, indicating that AtE activity plays a role in innate immune evasion. We further investigated this process during zebrafish infections using *E. faecalis* derivatives expressing the GFP and transgenic zebrafish with macrophages expressing mCherry (Fig 9). The ratio of GFP fluorescence intensity inside and outside of macrophages showed a significant increase in macrophage

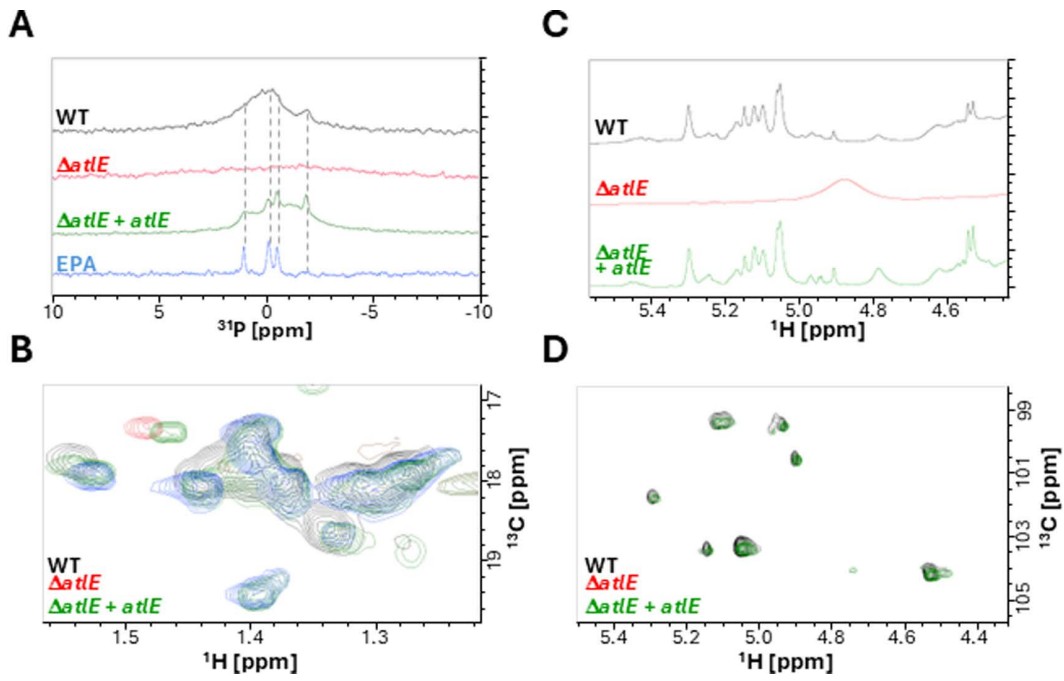

**Fig 8. NMR analysis of EPA released in culture supernatants.** *E. faecalis* cells were grown in chemically defined medium and filtered supernatants were concentrated, dialysed, freeze-dried and resuspended in $D_2O$ for NMR analysis. **A**, $^{31}P$ NMR spectra corresponding to OG1RF (WT, black), *atlE* mutant (Δ*atlE*, red), complemented *atlE* mutant (Δ*atlE* + *atlE*, green) and purified EPA (blue) samples. Dashed lines show the $^{31}P$ signals across samples. **B**, $^1H$-$^{13}C$ NMR HSQC showing the methyl rhamnose proton region. **C** and **D** show $^1H$ (1D) and $^1H$-$^{13}C$ (2D) HSQC spectra corresponding to the anomeric region.

uptake for the Δ*atlE* mutant as compared to the parental strain (Fig 9A and 9B). Macrophage evasion could be restored to parental levels upon complementation (Fig 9C and 9D).

We next investigated the contribution of cell wall fragments released by AtlE to *E. faecalis* virulence (Fig 10). The deletion of *atlE* led to a significant decrease in embryo lethality as compared to wild type cells and virulence could be complemented (Fig 10A and S8 Fig). The virulence of the mutant could also be restored when Δ*atlE* cells were co-injected with cell wall fragments (Fig 10B and S9 Fig). Whilst soluble fragments injected alone did not have any effect on the zebrafish embryos, they significantly increased lethality of OG1RF cells (Fig 10C and S10 Fig). By contrast, cell wall fragments from the *epaR* strain, which does not produce any EPA decorations [20], had no impact on OG1RF virulence, even when higher doses were injected (Fig 10D and S11 Fig). Collectively, these results therefore suggested that the release of EPA from the cell wall by AtlE contributes to *E. faecalis* pathogenesis.

## Discussion

Peptidoglycan hydrolases belong to multigene families, and their functional redundancy makes it difficult to identify their specific roles. *E. faecalis* strains encode at least fifteen potential peptidoglycan hydrolases, four of which have been previously characterized (AtlA [7], AtlB [6], AtlC [6] and EnpA [28]). All *E. faecalis* strains encode AtlA, the major peptidoglycan hydrolase responsible for complete daughter cell separation during cell division [7,11]. AtlB, AtlC and EnpA are part of prophage genomes and not encoded by all *E. faecalis* strains. Here, we identify AtlE, a peptidoglycan hydrolase which is also ubiquitous across *E. faecalis* strains. Interestingly, the muramidase activity of AtlE does not contribution towards the separation of daughter cells during exponential phase or stationary phase when AtlA is produced (Fig 3). In the absence

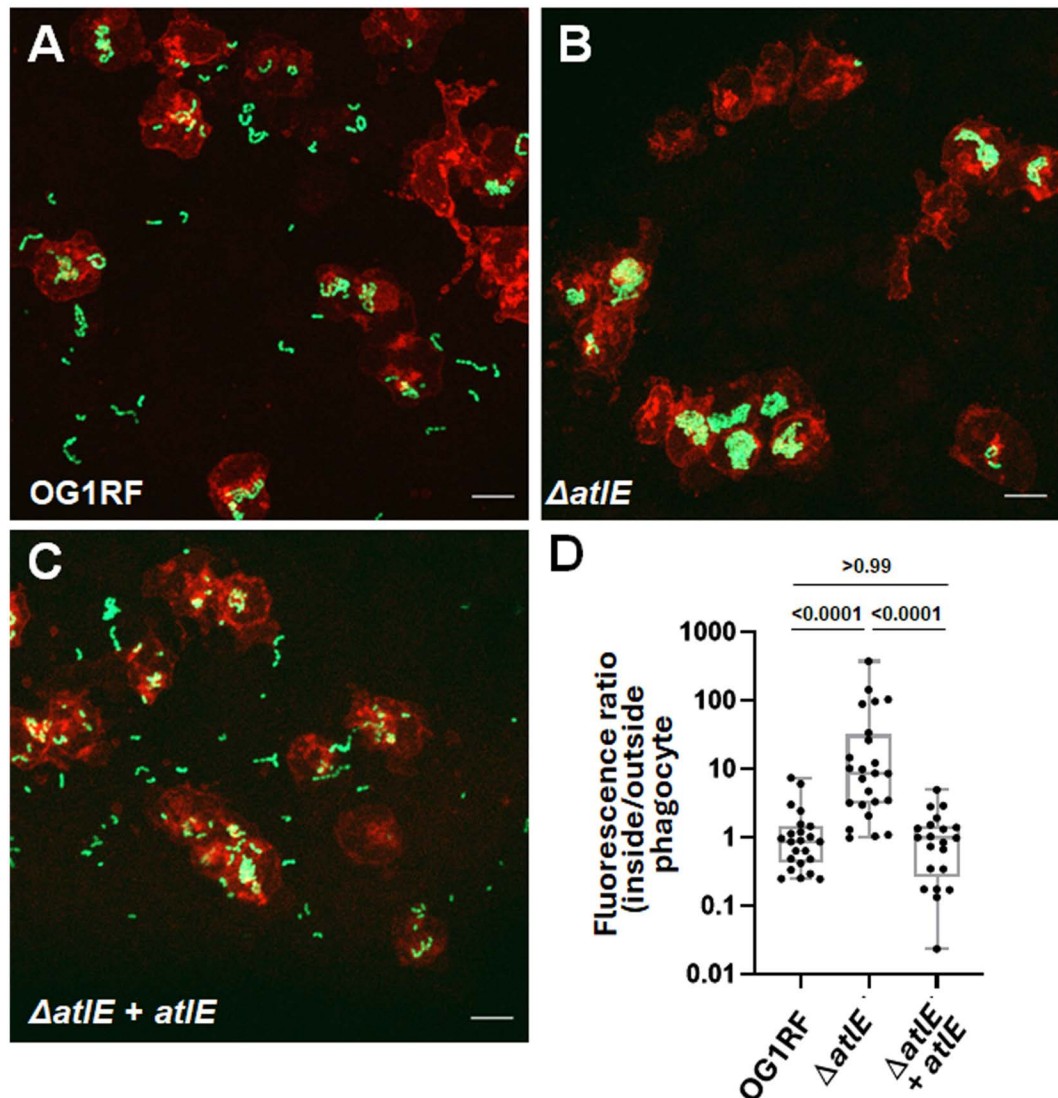

**Fig 9. Uptake of *E. faecalis* OG1RF, Δ*atlE* and complemented derivative by zebrafish macrophages.** Embryos of the *Tg*(*mpeg:mCherry-F*) trans-genic line were infected with c.a. 2,000 *E. faecalis* cells expressing GFP and fixed in 4% paraformaldehyde 1.5h post infection. Fluorescent bacteria and phagocytes were imaged by scanning confocal microscopy. The area of GFP fluorescence signal outside and inside macrophages was measured and the ratio of GFP fluorescence area inside to outside phagocytes was used to quantify bacterial uptake. Phagocytosis was significantly higher for the Δ*atlE* mutant when compared to the parental OG1RF or the complemented strain (P<0.0001 in both cases). No difference in uptake was found between the parental strain OG1RF and complemented mutant. Representative images of phagocytes following infection with wild type OG1RF **(A)**, Δ*atlE* (**B**) or complemented *atlE* mutant (Δ*atlE*+atlE, **C**) are shown. Macrophages appear in red, GFP-producing bacteria in green. Scale bar is 10 μm. **D**, Pairwise comparisons of fluorescence ratios. Statistical significance was determined by unpaired non-parametric Dunn's multiple comparison test.

of AtlA, AtlE can contribute to septum cleavage during stationary phase. Collectively, these results suggest that despite a limited overlapping role of these 2 enzymes, they fulfil different physiological roles. Our Western blot analysis suggests that both enzymes are produced during exponential growth, but AtlA is the most abundant of the two enzymes (Fig 4). This could (at least in part) explain the major contribution of AtlA to daughter cell separation. The decrease in abun-dance in AtlA is associated with a proteolytic cleavage of its N-terminal domain [11]. This cleavage has been proposed to impair the septal localisation of the enzyme and could potentially explain a preferential septum cleavage activity during

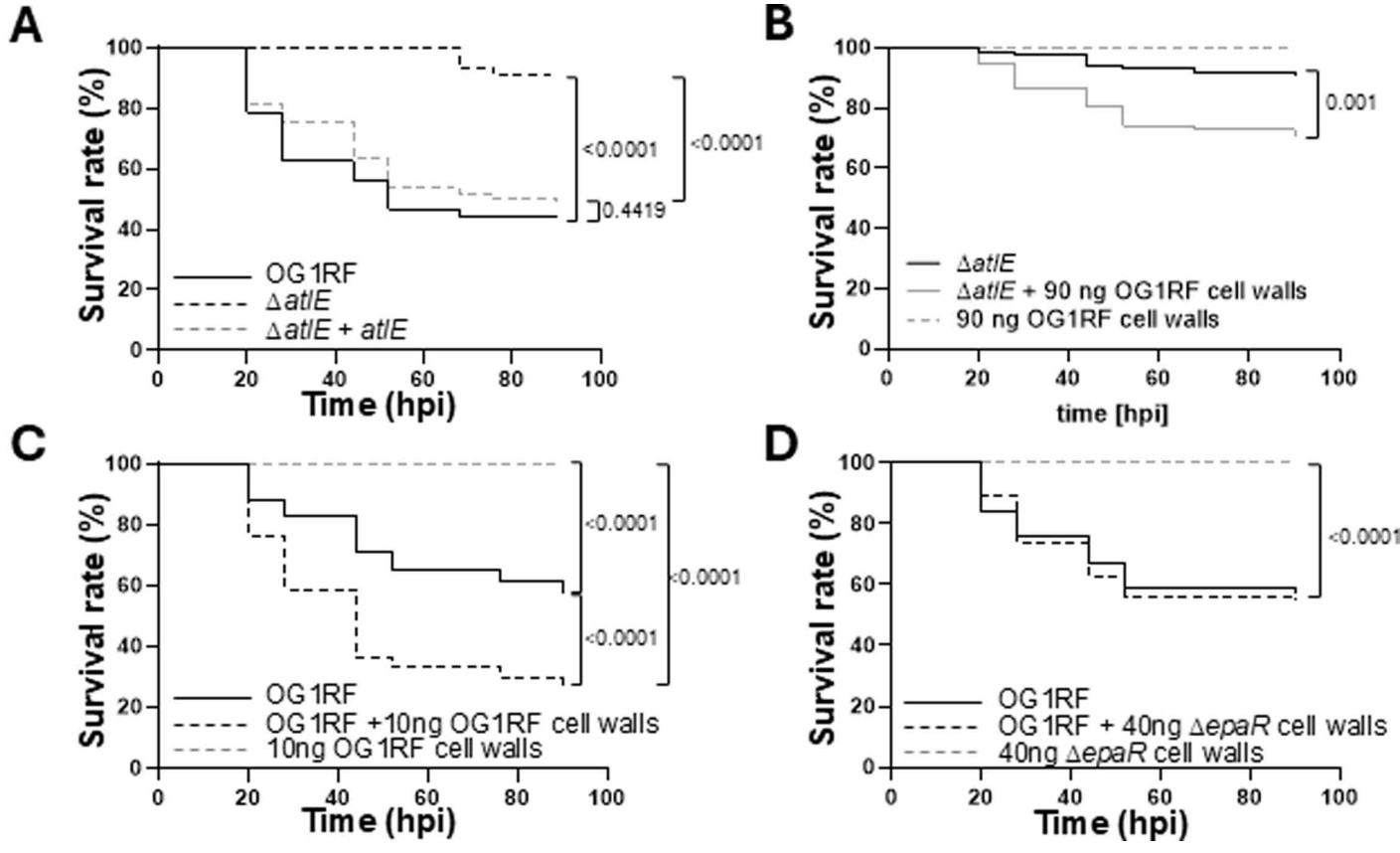

**Fig 10. Contribution of AtlE activity to *E. faecalis* virulence. A**, Survival of zebrafish larvae following injection with *c.a.* 2,000 CFUs of strains OG1RF, Δ*atlE* and complemented derivative (Δ*atlE* + *atlE*). **B**, Injection of Δ*atlE* in the presence of 90 ng of OG1RF soluble cell wall fragments. **C**, Injection of OG1RF in the presence of 10 ng of OG1RF soluble cell wall fragments. **D**, Injection of OG1RF in the presence of 40 ng of Δ*epaR* soluble cell wall fragments. Each panel combines three independent experiments, each corresponding to the injection of ≥25 embryos (individual repeats are shown in S8–S11 Figs).

exponential phase [29]. Transcriptional regulation has been reported in other bacteria, where sigma factors regulate the expression of peptidoglycan hydrolases in a growth phase-dependent manner [30,31]. Both AtlA and AtlE seem to be constitutively produced but how the genes encoding these two peptidoglycan hydrolases are regulated awaits further analysis.

AtlA activity is controlled by several post-translational regulations [8,11]. This work revealed this could also be the case for AtlE. The activity of AtlE requires the presence of strain-specific EPA decorations. It is therefore possible that the diversity in the C-terminal sequence and domain organisation of AtlE is driven by the structural diversity of decorations. Interestingly, strains V583 and OG1RF produce distinct EPA polysaccharides [20,21] but encode AtlE alleles with a DUF5776 domain at their C-terminus. This suggests that this family of domains has evolved to recognize different carbohydrate motifs but the binding activity of DUF5776 domains remains to be formally established. HR-MAS NMR and thin-section electron microscopy did not reveal any change in EPA structure, EPA surface exposure or cell envelope ultrastructure following the deletion of *atlE*. We previously showed that cell wall material can be released into supernatants during stationary phase [6]. The work carried out with *Staphylococcus aureus* indicated that soluble peptidoglycan fragments are unable to enhance the virulence of this pathogen [32] We therefore hypothesized that AtlE may be responsible for the release

of EPA in culture supernatants and explored whether this may contribute towards *E. faecalis* virulence in the zebrafish model of infection. The experimental data described in this work support this hypothesis. AtlE is critical for *E. faecalis* virulence, and the activity of this peptidoglycan hydrolase underpins the release of EPA in culture supernatants (Fig 8) and phagocyte evasion (Fig 9).Since the injection of wild-type enterococci in the presence of soluble cell wall fragments with intact EPA decorations significantly increases the killing of zebrafish embryos, this study elucidated the role of AtlE during pathogenesis. Collectively, our results describe a novel virulence mechanism whereby peptidoglycan remodelling mediated by AtlE contributes to the release of EPA fragments as decoy molecules that can inhibit the host immune response. Similar strategies have been described in other pathogens. In *Bordetella pertussis*, the release of surface β-1,6-*N*-acetyl-D-glucosamine polysaccharides promotes respiratory tract colonization by resisting killing by antimicrobial peptides [33]. Additionally the outer membrane vesicles released by *Acinetobacter baumanii* also contribute to resisting antimicrobial peptides and promote virulence [34], and in *E. coli*, the release of LPS has been proposed to trigger excessive accumulation of systemic ATP, leading to impaired polymorphonuclear leukocyte chemotaxis [35]. The mechanism by which EPA fragments interfere with the immune response awaits further investigation to understand how this complex surface polysaccharide modulates immunity. The discovery of AtlE and the contribution to cell wall remodelling reveals another strategy evolved by *E. faecalis* to thrive in the host.

## Materials and methods

### Ethics statement

The Jagiellonian University Zebrafish Core Facility (ZCF) is a licensed breeding and research facility (District Veterinary Inspectorate in Krakow registry; Ministry of Science and Higher Education record no. 022 and 0057).

All larval zebrafish experiments were conducted in accordance with the European Community Council Directive 2010/63/EU for the Care and Use of Laboratory Animals of Sept. 22, 2010 (Chapter 1, Article 1 no.3) and Poland's National Journal of Law act of Jan. 15, 2015, for Protection of animals used for scientific or educational purposes (Chapter 1, Article 2 no.1). All experiments with zebrafish were done on larvae up to 5 days post fertilization, which have not yet reached the free feeding stage, and were performed in compliance with ARRIVE guidelines.

**Bacterial strains, plasmids, and growth conditions.** All strains and plasmids used in this study are described in Table 1. The bacteria were grown at 37°C in Brain Heart Infusion broth or agar (15 g/L) (BHI, Difco laboratories, Detroit, USA). When required, *E. coli* was grown in the presence of 100 µg/mL ampicillin (for protein expression) or 200 µg/mL erythromycin (for pGhost selection). *E. faecalis* transformants were selected with 30 µg/mL erythromycin to select for pGHost9 and pTet derivatives. For complementation experiments, anhydrotetracycline was used at a concentration of 10 ng/µL for precultures and 50 ng/µL unless stated otherwise.

### Plasmid construction

All plasmids and oligonucleotides are described in S1 Table. pGhost derivatives for gene deletion were constructed using the same strategy. Two homology regions flanking the open reading frame encoding each peptidoglycan hydrolase (EF0252, EF0114 and AtlE) were amplified from genomic DNA via PCR. The 5' arm (~0.75 kb) was amplified using the primers H11 (sense) and H12 (antisense), whereas the 3' arm (~0.75 kb) was amplified using H21 (sense) and H22 (antisense). Once purified, the two PCR products were mixed (equimolar amount of each) and fused into a single product (~1.5 kb) via splice overlap extension PCR [36] using primers H11 and H22. The resulting fragment was cut with XhoI and EcoRI and cloned into pGhost9 vector [37] cut with the same enzymes. Candidate pGhost derivatives were screened by PCR using primers pGhost_Fw and pGhost_Rev. A positive clone containing the fused H1-H2 insert corresponding to each construct was checked by Sanger sequencing. pGHH_0252, pGHH_0114, pGHH_atlE_J were used to build mutants in the JH2–2 genetic background; pGHH_atlE_O was used to build an *atlE* mutant in the OG1RF genetic background.

pET_AtlE_O was built to express AtlE encoded by OG1RF. The DNA fragment encoding amino acids 25 to 818 was PCR amplified from OG1RF genomic DNA with oligonucleotides atlE_O_pETF and atlE_O_pETR using Phusion DNA polymerase (Fisher Scientific). The resulting fragment was cloned in frame with the hexahistidine sequence of pET2818, a pET2816b derivative [38], using XbaI and BamHI.

*atlE* deletion mutants in JH2-2 and OG1RF backgrounds were complemented using pTetH derivatives, allowing anhydrotetracycline inducible expression. DNA fragments encoding the full length AtlE were PCR amplified using atlE_J_Fw and atlE_J_Fw (JH2-2 allele) or atlE_O_Fw and atlE_O_Fw (OG1RF allele). PCR fragments were digested with XbaI and BamHI and cloned into pTetH similarly digested. Positive clones were screened using primers pGhost_Fw and pGhost_Rev. A positive clone containing the *atlE* insert was checked by sanger sequencing and complementation plasmids were named pTet_AtlE_J and pTet_AtlE_O.

**Construction of *E. faecalis* mutants by allele exchange.** The protocol described previously was followed [6]. *E. faecalis* was electroporated with pGhost9 derivatives and transformants were selected at 30°C in the presence of erythromycin. Single crossing-overs were induced at non permissive temperature (42°C) and screened by PCR. The second recombination event was triggered by subculturing recombinant clones in BHI at 42°C. Erythromycin sensitive colonies were screened by PCR to identify mutants using flanking oligos (H110 and H220).

**Production and purification of recombinant proteins.** *E. coli* Lemo21(DE3) cells (NEB) transformed with pET_AtlE_O was grown in ZY auto-induction medium [39] containing 100 µg/mL of ampicillin and 35 mg/mL of chloramphenicol at 37 °C until the $OD_{600}$ reached 0.5. The cultures were cooled on ice to room temperature and incubated overnight at 20 °C. Cells were harvested by centrifugation, and the frozen pellets were resuspended in lysis buffer (50 mM Tris-HCl pH 8, 300 mM NaCl, 10 mM imidazole). After a 30-minute incubation with 0.1 mg/mL deoxyribonuclease and 0.02M magnesium sulphate at 4°C, the cells were disrupted by sonication on ice. The lysate was clarified by centrifugation, and the supernatant was applied onto a 5mL nickel-chelate affinity prepacked column using an ÄKTA Pure system (Cytiva). The resin was washed with 50 mM Tris-HCl pH 8, 300 mM NaCl, and 5 mM imidazole. The protein was eluted with a gradient to 50 mM Tris-HCl pH 8, 300 mM NaCl, and 500 mM imidazole. The protein was further purified on a Superdex 200 16/60 column (Cytiva) equilibrated with 20 mM Tris-HCl pH 7.5, 150 mM NaCl. Fractions containing pure AtlE (>90%), as analyzed by SDS-PAGE, were pooled, concentrated to 2.5 mg/mL, and stored at -80°C in the presence of 25% glycerol until further use.

## Preparation of *E. faecalis* crude extracts

Crude extracts were prepared from cell pellets corresponding to the equivalent of 40 ml at $OD_{600nm}$ ~1) washed in PBS. Cells were resuspended in 750 µl of PBS and transferred to a tube containing 250 µl of glass beads (100 µm diameter, Sigma) and mechanically disrupted using a FastPrep device (six cycles of 30 s at maximum speed with 2 min pauses between cycles). Protein concentration was determined using a BioRad protein assay.

## Detection of peptidoglycan hydrolytic activities by zymogram

Proteins were separated by SDS-PAGE using gels containing autoclaved *E. faecalis* JH2–2 Δ*ponA* Δ*pbpZ* Δ*pbpF*, a mutant harbouring a triple deletion of class A PBPs genes. After electrophoresis, proteins were renatured by incubating the gel in 25 mM Tris–HCl (pH 7.5) containing 0.1% Triton at 37 °C. Gels were stained with 1% (w/v) methylene blue in 0.01% (w/v) potassium hydroxide.

## Western blot analyses

Proteins were separated on a 11% SDS-PAGE and transferred to a nitrocellulose membrane. After a blocking step for 1h at room temperature in Tris buffer saline (TBS, 10 mM Tris-HCl pH7.5, 150 mM NaCl) supplemented with 0.025% tween-20 v/v) and 2% skimmed milk (w/v), the membrane was incubated with rabbit polyclonal anti-AtlA antibodies raised

against the full-length AtlA (residues 54–737) (1:25,000 dilution) or rabbit polyclonal anti-AtlA antibodies raised against the C-terminal domain of AtlE (residues 391–818) (1:10,000 dilution) Proteins were detected using goat polyclonal anti-rabbit antibodies conjugated to horseradish peroxidase (Sigma) at a dilution of 1:20,000 and clarity Western ECL Blotting Substrate (BioRad).

## Cell wall extractions

Overnight static cultures at 37°C were diluted 1:100 into 900 mL of fresh broth ($OD_{600}$ ~0.02) and grown without agitation to an $OD_{600}$ ~1. Cells were harvested by centrifugation (10,000 x $g$, room temperature for 10 min) and resuspended in boiling MilliQ water. SDS was added to a final concentration of 5% (w/v). After 30 min at 100°C, cell walls were recovered by centrifugation (5 min at 15,000 x $g$, room temperature) and washed 5 times with MilliQ water.

## Muropeptide analysis using rp-HPLC and LC-MS/MS

Cell walls (1 mg) were digested in a final volume of 100 µL for 16h at 37°C in the presence of 100U of mutanolysin (SIGMA) or 250 µg of recombinant AtlE in 10 mM phosphate buffer (pH5.5) or 25 mM Tris-HCl (pH7.5), respectively. Soluble cell wall fragments were recovered by centrifugation and reduced by addition of one volume of 200 mM borate buffer (pH 9) and 500 µg of sodium borohydride. The pH was adjusted to 4.5 with phosphoric acid and muropeptides were analysed by rp-HPLC and LC-MS/MS.

Rp-HPLC UV traces shown in Fig 6 were acquired on a Dionex Ultimate 3000 system operated at 0.3 mL/min. Samples corresponding to the digestion of 200 µg of cell walls were injected on a Hypersil Gold aQ column (2.1 mm x 200 mm, 1.9 µm particles; Thermo Fisher) using water + 0.1% formic acid (v/v) as buffer A and acetonitrile + 0.1% formic acid (v/v) as buffer B. After 1 column volume (CV) at 0% B, muropeptides were eluted with a 12 CV gradient to 20% B. The column was washed with 3 CV of 95% B and equilibrated in buffer A.

LC-MS/MS analysis was carried out on Ultimate 3000 RSLCnano system (Dionex) coupled to an LTQ-Orbitrap Elite Hybrid mass spectrometer (Thermo Fisher Scientific) equipped with an EASY-Spray ion source. Samples corresponding to 100 ng of cell wall digestion were separated using an EASY-Spray C18 capillary column (150 µm × 150 mm, 2-µm particle size; PN ES806, Thermo Fisher). Muropeptides were eluted using a water-acetonitrile + 0.1% formic acid (v/v) at a flow rate of 1.5 µL/min. The mass spectrometer operated in a standard data-dependent acquisition mode controlled by the Xcalibur v.3.0 and LTQ Tune Plus v.2.7 software. The instrument was operated with a cycle of one MS (in Orbitrap) acquired at a resolution of 60,000 from an $m/z$ range of 100–1,600. The top 10 most abundant multiply charged ions (2+ and higher) were subjected to higher-energy collisional dissociation (HCD) fragmentation (normalized collision energy = 30, activation time = 0.1 ms).

## Genomic analysis

To investigate the occurrence and diversity of AtlE among *E. faecalis*, we downloaded 426 genomes deposited in the Joint Genome Institute's Integrated Microbial Genomes (IMG/JGI) database. The conserved N-terminal domain containing GH25 of the model bacterium *E. faecalis* V583 was used as a query for BLASTP (stringency e-80). The top hit from each genome was obtained and ORFs were aligned using MUSCLE. Sequences were trimmed to remove amino acids outside the GH25 domain. Phylogenetic reconstruction was performed using IQTree v1.6.8 performed using 1000 bootstraps [40]. Tree visualisation and subsequent analysis was performed using the Interactive Tree of Life (ITOL) server [41]. To determine C-terminal identity, BLAST P using either V583 (DUF5776) or JH2–2 (GW) as the queries were performed.

## LC-MS/MS identification of *E. faecalis* proteins in culture supernatants

*E. faecalis* JH2–2 Δ*atlABC* cells were grown overnight in 50 mL of BHI. Following centrifugation for 10 min at 15,000 x $g$, the supernatant was filtered (0.45 µm pores). Proteins were precipitated by addition of TCA 10% (w/v) final. After 10 min

on ice, the sample was spun at 10,000 x $g$ for 15 min and the supernatant was discarded. Proteins were washed in 20 mL acetone, left to dry under a fume cupboard and resuspended in 250 µL of SDS-PAGE loading buffer.

Following Coomassie staining, the gel was washed twice with MilliQ water for 10 min each. The bands of interest were cut out and incubated in 100 µL of a 100 mM ammonium bicarbonate solution for 10 min. The buffer was then removed and replaced by 50 µL of a mixture containing 25 mM ammonium bicarbonate ($NH_4HCO_3$) in acetonitrile (ACN). The procedure was repeated twice until all Coomassie stain was removed and the gel slices were resuspended in 100% ACN for 30 min at 37°C. ACN was then removed and replaced with 50 µL of 10 mM DTT in 100 mM $NH_4HCO_3$.

Reduction was carried out for 45 min at 55°C with agitation, and the buffer was replaced with 50 µL of 50 mM of iodoacetamide in 100 mM $NH_4HCO_3$. Alkylation was carried out in the dark for 30 min at room temperature with agitation, and gel slices were washed with 100 µL of 100 mM $NH_4HCO_3$ for 5 min at room temperature with agitation followed by 100 µL of 100% ACN for 15 min at room temperature with agitation. Gel pieces were air dried for 5–10 minutes and digested in 50 µL of 0.01 mg/mL trypsin solution for 4h at 37°C. Formic acid was added to digestion products at a final concentration of 5% to stop the enzymatic reaction. The solution was transferred to an Eppendorf tube, and peptides were further extracted from the gel by adding 100 µL of 1% (v/v) formic acid and incubating for 15 min at RT with shaking followed by the addition of 100 µL of 100% ACN for 15 min at RT with shaking. This procedure was repeated once, and the combined peptide solutions were dried under vacuum. Peptides were resuspended in 10 µL of 0.1% (v/v) formic acid and cleaned up using a zip-tip (Merck).

## High-resolution magic angle spinning NMR

Cells were grown in BHI broth overnight and this starter culture was then used to inoculate 100 mL of BHI broth at a starting $OD_{600}$ of 0.04 until an $OD_{600}$ reached 0.7 was reached. Cells were then heat-killed at 60 °C for 20 minutes. The suspension was then washed twice in $D_2O$ and freeze-dried. Cells were then re-suspended into 100 µL of $D_2O$ containing 0.01% (v/v) acetone as an internal standard for chemical shifts (δ $^1H$ 2.225 and δ $^{13}C$ 31.55) and centrifuged at 3,000 rpm to be packed into 4-mm $ZrO_2$ rotor (CortecNet, Paris, France) [42].

High-resolution magic angle spinning NMR was performed on *E. faecalis* OG1RF and isogenic mutant Δ*11720* strain using an 18.8T advance NEO spectrometer where $^1H$ and $^{13}C$ resonated at 800.12 and 200.3 MHz respectively. The set of pulse programs used was extracted from the Bruker pulse program library where pulses (both hard and soft pulses and their powers) and delays were optimized for each experiment. $^1H$-$^{13}C$ HSQC spectra were recorded with an inept sequence to distinguish secondary carbons and carbons bearing primary alcohols from other carbons. The 18.8 T was equipped with a 4mm D/$^1H$/$^{13}C$/$^{31}P$ HR-MAS probe head where the rotor was spun at 8000Hz during acquisition to eliminate anisotropy effect of jelly state of bacterial cells. All spectra were recorded at 300 K, and the rotor spinning rate was 8 kHz. For $^1H$-$^{13}C$ HSQC experiments, the spectral widths were 12,820 Hz ($^1H$) with 1,024 points for the FID resolution and 29,994 Hz ($^{13}C$) with 400 points for FID resolution during 400 scans, giving 12.5 Hz/pt and 75.0 Hz/pt, respectively.

## NMR analysis of culture supernatants

A preculture was grown in THY (Todd Hewitt, 30 g/L, Yeast extract 10 g/L) to an $OD_{600nm}$ of 0.5 at 37°C and used to inoculate 2L of chemically defined medium with the following composition (per L): 100 mg of each L- aminoacid; 10 mg of adenine, uracil, xanthine and guanine, 200 mg $MgCl_2$-6$H_2O$, 50 mg $CaCl_2$-2$H_2O$, 5 mg $FeCl_3$-4$H_2O$, 5 mg $ZnSO_4$-7$H_2O$, 2.5 mg $CoCl_2$-6$H_2O$, 0.1 mg $CuSO_4$-5$H_2O$, 30 mg $MnSO_4$-$H_2O$, 2.72 g $KH_2PO_4$, 10.44 g $K_2HPO_4$, 0.6 g ammonium citrate, 1 g soium citrate, 10 g glucose, 2 mg pyridoxal-HCl, 1 mg nicotinic acid, 1 mg thiamine-HCl, 1 mg riboflavin, 1 mg panthotenic acid, 10 mg para-aminobenzoic acid, 1 mg D-biotin, 1 mg folic acid, 1 mg vitamin B12, 5 mg orotic acid, 5 mg thymidine, 5 mg inosine, 2.5 mg thioctic acid and 5 mg pyridoxamine-HCl. After 24h at 37°C, cells were spun at 6,000 rpm for 15 minutes and supernatants were passed through a 0.22 µm filter. Filtered supernatants were concentrated to 50 mL using a Millipore stirred cell with a 10 kDa MWCO membrane, dialysed thrice against 2 L of MilliQ water. Dialysed supernatants were

then freeze-dried and resuspended in 600 µL of $D_2O$ and 0.01% acetone as a standard. NMR experiments were run on a Bruker 500 MHz (1D) and 600 MHz (2D) Neo. Spectra were processed using Topspin 4.5.0 Experiments were run with the following number of scans: for $^1H$-$^{13}C$ HSQC, 8 scans and 16 dummy scans; for $^1H$ 1D sectra, 16 scans and 4 dummy scans and for $^{31}P$ 1D spectra, 256 scans and 8 dummy scans.

## Flow cytometry

Overnight static cultures at 37°C were diluted 1:100 into fresh broth ($OD_{600}$ ~ 0.02) and grown to mid-exponential phase (OD600 ~ 0.2 to 0.4). Bacteria were diluted 1:100 in filtered phosphate buffer saline and analyzed by flow cytometry using a Millipore Guava easyCyte H2L system and the GuavaSoft v3.1.1 software. Light scatter data were obtained with logarithmic amplifiers for 20,000 events.

## Chemical fixation, thin sectioning, and electron microscopy of *E. faecalis* cells

Cell pellets were fixed in 3% (w/v) glutaraldehyde at 4 °C overnight. Samples were washed in 0.1 M sodium cacodylate buffer and incubated for 2 h at room temperature in 1% (w/v) osmium tetroxide for secondary fixation. Cell pellets were washed with 0.1 M sodium cacodylate buffer and dehydrated by incubating with increasing concentrations of ethanol (50% (v/v), 75% (v/v), 95% (v/v), 100% (v/v) ethanol) for 15 min each. Ethanol was removed and samples were incubated with propylene oxide for complete dehydration. Samples were incubated overnight at room temperature in a 1:1 mix of propylene oxide and Araldite resin (Agar Scientific, CY212) to allow for infiltration. Resin was removed and excess propylene oxide evaporated at room temperature. Samples were placed in two consecutives 4 h incubations in pure Araldite resin before being embedded into the final fresh resin. Resin was polymerised by incubation at 60 °C for 48 h. Thin sections (90 nm) were produced using an Ultracut E Ultramicrotome (Reichert-Jung) and floated onto 300-square mesh nickel TEM grids. Sections were stained in 3% (w/v) uranyl acetate for 30 min, washed with $dH_2O$, stained with Reynold's lead citrate for 5 min and further washed with $dH_2O$. Sections were imaged on a FEI Tecnai T12 Spirit Transmission Electron Microscope operated at 80 kV and equipped with a Gatan Orius SC1000B CCD camera.

## Zebrafish survival experiments

Zebrafish embryos were obtained by the natural spawning of adult zebrafish (line AB/TL or *tg(mpeg:mcherry-F* [43]), which were housed in a continuous recirculating closed-system aquarium with a light/dark cycle of 14/10 h at 28 °C. Dechorionated embryos at 30 hpf were anaesthetized and systemically injected with ca. 2,000 *E. faecalis* cells as previously described [18]. In some experiments, the bacterial inoculum was supplemented with soluble cell wall fragments dissolved in MilliQ water. The number of injected cells was checked before and after each series of injections with a given strain. The infected embryos were monitored at regular intervals until 90 hpi (hpi). At least 25 embryos per group were used in each experiment.. For phagocyte-depletion, the *pu.1* knockdown was performed as described previously [18].

To prepare cell walls fragments for zebrafish exoeriments, OG1RF and its isogenic Δ*epaR* mutant (producing EPA lacking decorations; [20]) were grown overnight for 16h in 500 mL of Brain Heart Infusion broth at 37°C. Cells were harvested and boiled in 4% (w/v) in a volume of 40 mL for 30 minutes and washed five times in MilliQ water. After a pronase treatment (4 hours at 50°C in 2 mL of 20 mM Tris-HCl containing 2 mg/mL of pronase), SDS was added to a final concentration of 1%. Pronase was inactivated for 15 minutes at 100°C and the cell walls were washed six times in MilliQ water and freeze-dried. Ten mg of cell walls were digested in the presence of 150 units of mutanolysin (SIGMA) in a final volume of 250 µL of 10 mM phosphate buffer at pH 5.5. Soluble cell wall fragments were recovered after centrifugation and freeze dried. Cell wall fragments were weighed once freeze-dried. The yield of soluble fragments was similar for OG1RF and the Δ*epaR* mutant (3.5 and 3.7 mg, respectively).

**Imaging of infected larvae by confocal microscopy and quantification of bacterial uptake by phagocytes**

Larvae were fixed in 4% (w/v) paraformaldehyde at 1.5 hpi. Fixed larvae washed with PBS were then immersed in 1% (w/v) low-melting-point agarose solution in E3 medium and mounted flat on a glass-bottomed dish. Images were acquired with a Zeiss LSM900 Airyscan 2 confocal laser scanning microscope using the C-Apochromat 40x/NA 1.2 water objective. Maximum projections were used for representative images. No non-linear normalisation was performed. Bacterial phagocytosis was quantified as previously described [11]. Briefly, all bacterial clusters were identified based on their GFP fluorescence. Next, the fluorescence intensities of mCherry-labelled macrophages surrounding the bacteria (2 µm radius) were analysed using a custom ImageJ script called Fish Analysis v5 (http://sites.imagej.net/Willemsejj/). The phagocytosed bacteria had a high intensity of mCherry fluorescence (inside the macrophages) in the surrounding area, and the cut-off of 2 times the background level was used to distinguish the phagocytosed from non-phagocytosed bacteria. The area of phagocytosed bacteria was compared to the area of non-phagocytosed bacteria and their ratio was calculated.

**Statistical analyses**

Statistical analyses were performed using GraphPad Prism. Survival experiments were evaluated using the Kaplan-Meier method. Comparisons between curves were made using the Log Rank (Mantel-Cox) test. For flow cytometry experiments, forward scattered light values were compared using a one-way ANOVA (Tukey's multiple comparison test). For macrophage uptake experiments, fluorescence intensity ratios were compared using an unpaired non-parametric Dunn's multiple comparison test.

**Supporting information**

**S1 Fig. Zymogram analysis of *E. faecalis* JH2–2 culture supernatants.** Peptidoglycan hydrolytic activities were detected in 20 µL of culture supernatants of strains JH2–2 (WT), Δ*atlABC*, Δ*atlABC Δ0252* and Δ*atlABC Δ0114* grown overnight. Cells from the triple class A PBP mutant Δ*ponA ΔpbpF ΔpbpZ* were used as a substrate and zymograms were incubated for 72h at 37°C.
(TIF)

**S2 Fig. Comparison of OG1RF and JH2–2 loci encoding EPA.** Genes *epaA* to *epaR* are conserved across strains *epaR* to *11706* encode EPA decorations which can vary between strains.
(TIF)

**S3 Fig. Zymogram analysis of *E. faecalis* OG1RF culture supernatants.** Peptidoglycan hydrolytic activities were detected in 25 µL of culture supernatants of strains OG1RF (WT, lane 1), Δ*atlE (lane 2),* Δ*atlA* (lane 3), Δ*atlA* Δ*atlE* (lane 4), and complemented Δ*atlA* Δ*atlE* mutant (Δ*atlE + atlE*, lane 5). Cells from the triple class A PBP mutant Δ*ponA ΔpbpF ΔpbpZ* were used as a substrate.
(TIF)

**S4 Fig. MS/MS analysis of peak 5. A,** fragmentation of the doubly charged Ion $(M+2H]^{2+}$; $m/z=555.78$). Ions with $m/z$ values matching predicted fragments are boxed in red. **B,** List of predicted fragments, theoretical and observed $m/z$. ND, not detected; g, GlcNAc; m(r), reduced MurNAc; residues in square bracket correspond to the lateral chain.
(TIF)

**S5 Fig. Specificity and sensitivity of antibodies recognizing AtlA and AtlE. A,** Specificity of antibodies raised against recombinant AtlA and AtlE proteins was tested against *E. faecalis* crude extracts from cells grown in exponential phase ($OD_{600nm} \approx 0.3$); WT is OG1RF, Δ*atlE + E* corresponds to the Δ*atlE* mutant complemented. For AtlA detection, 2 µg of crude extracts were used; primary serum was used at a dilution of 1/25,000. For AtlE detection, 5µg of crude extracts were used;

primary serum was used at a dilution of 1/10,000. In both cases, secondary antibodies (goat anti-rabbit antibodies coupled to horseradish peroxidase) were used at a 1/20,000 dilution. **B**, sensitivity of anti-AtlA and anti-AtlE antibodies.
(TIF)

**S6 Fig. AtlE is not involved in EPA biosynthesis or exposure at the cell surface. A**, $^1$H-$^{13}$C HSQC spectra of purified wild type and $\Delta atlE$ EPA. The region displayed corresponding to the anomeric protons (4.3-5.5 ppm) and anomeric carbons (90–110 ppm) of WT (left) and $\Delta atlE$ (right) did not reveal any major difference between the 2 EPA polymers. **B**, $^1$H-$^{13}$C HSQC HR-MAS NMR experiments recorded on *E. faecalis* OG1RF (left) and $\Delta atlE$ (right) cells show that AtlE does not contribute towards the production or display of surface exposed EPA or lipoteichoic acid (LTA) [26]. Two other currently unidentified cell wall polysaccharides denoted with an asterisk are also detected [21]. **C**, Thin section transmission electron microscopy of *E. faecalis* OG1RF (left) and $\Delta atlE$ (right) cells confirm EPA decorations remain surface exposed in the $\Delta atlE$ mutant, both forming a pellicle at their cell surface (arrows).
(TIF)

**S7 Fig. Survival ratio of WT and phagocyte-depleted zebrafish larvae infected with *E. faecalis* OG1RF, $\Delta atlE$ and $\Delta atlE$ complemented strains.** Larvae were infected with *ca.* 1,650 CFUs of parental OG1RF strain (solid red line) or $\Delta atlE$ (solid black line). Phagocyte depletion was performed using *pu.1* morpholinos before injection with OG1RF (green dashed line), $\Delta atlE$ (solid blue line) or $\Delta atlE + atlE$ cells (green solid line). Survival was monitored between 20–90 hours post infection (hpi) at 28°C using 25 larvae per strain.
(TIF)

**S8 Fig. Survival ratio of zebrafish larvae infected with *E. faecalis* OG1RF, $\Delta atlE$ and $\Delta atlE$ complemented strains.** Larvae were infected with *ca.* 2,000 CFUs of parental (WT) OG1RF strain (solid line), $\Delta atlE$ (black dashed line) or $\Delta atlE + atlE$ (grey dashed line). Survival was monitored between 20–90 hours post infection (hpi) at 28°C using 25 larvae per strain per experiment. Three independent experiments (**A**, **B** and **C**) and combined results (**D**) are shown. (**E**) *P* values of pairwise comparison.
(TIF)

**S9 Fig. Survival ratio of zebrafish larvae infected with *E. faecalis* $\Delta atlE$ in the presence or absence of OG1RF soluble cell wall fragments.** Larvae were infected with *ca.* 2,000 CFUs of the $\Delta atlE$ strain in the absence (solid line) or presence (grey line) of 90 ng of soluble cell walls. A control injection corresponding to 90 ng of OG1RF cell walls alone is shown (grey dashed line). Survival was monitored between 20–90 hours post infection (hpi) at 28°C using at least 25 larvae per strain per experiment. Three independent experiments (**A**, **B** and **C**) and combined results (**D**) are shown. **E**, *P* values of pairwise comparisons.
(TIF)

**S10 Fig. Survival ratio of zebrafish larvae infected with *E. faecalis* OG1RF in the presence or absence of OG1RF soluble cell wall fragments.** Larvae were infected with *ca.* 2,000 CFUs of parental (WT) OG1RF strain in the absence (solid line) or presence (black dashed line) of 10 ng of soluble cell walls. A control injection corresponding to 10ng of OG1RF cell walls alone is shown (grey dashed line). Survival was monitored between 20–90 hours post infection (hpi) at 28°C using 25 larvae per strain per experiment. Three independent experiments (**A**, **B** and **C**) and combined results (**D**) are shown. (**E**) *P* values of pairwise comparison.
(TIF)

**S11 Fig. Survival ratio of zebrafish larvae infected with *E. faecalis* OG1RF in the presence or absence of *epaR* soluble cell wall fragments lacking EPA decorations.** Larvae were infected with *ca.* 2,000 CFUs of parental (WT) OG1RF strain in the absence (solid line) or presence (black dashed line) of 40 ng of soluble cell walls. A control injection

corresponding to 40 ng of OG1RF cell walls alone is shown (grey dashed line). Survival was monitored between 20–90 hours post infection (hpi) at 28°C using 25 larvae per strain per experiment. Three independent experiments (**A**, **B** and **C**) and combined results (**D**) are shown. (**E**) *P* values of pairwise comparison.
(TIF)

**S1 Table. Strains, plasmids, oligonucleotides.**
(DOCX)

**S1 File. Raw data.** Zipped raw data used to make figures 1–10 are provided in individual folders for each figure.
(ZIP)

**S2 File. Phosphorus 1D NMR.** 1000: OG1RF (WT sample) 1D phosphorus NMR. 1002: Δ*atlE* (mutant sample) 1D phosphorus NMR. 1003: Δ*atlE* + *atlE* (complemented mutant sample) 1D phosphorus NMR.
(RAR)

**S3 File. Proton 1D NMR.** 5000: OG1RF (WT sample) 1D proton NMR. 5002: Δ*atlE* (mutant sample) 1D proton NMR. 5004: Δ*atlE* + *atlE* (complemented mutant sample) 1D proton NMR.
(RAR)

**S4 File. HSQC 2D NMR.** 5001: OG1RF (WT sample) $^1$H-$^{13}$C HSQC NMR. 5003: Δ*atlE* (mutant sample) $^1$H-$^{13}$C HSQC NMR. 5005: Δ*atlE* + *atlE* (complemented mutant sample) $^1$H-$^{13}$C HSQC NMR.
(RAR)

**S1 Striking Fig. The activity of the peptidoglycan hydrolase AtlE releasing Enterococcal Polysaccharide Antigen (EPA) from cell surface is critical for phagocyte evasion.** Zebrafish embryos of the *Tg*(*mpeg:mCherry-F*) transgenic line were infected with 2,000 CFUs of *E. faecalis* cells constitutively producing GFP. Representative images show *E. faecalis* uptake in zebrafish embryos 1.5 h post infection with the WT strain (OG1RF) and a Δ*atlE* derivative. Phagocytes labeled with mCherry appear in red, and GFP-labelled bacteria in green. Scale bar is 10 µm. Robert E Smith, Bartosz J Michno, Rene L Christena, Finn O'Dea, Jessica L Davis, Ian D.E.A. Lidbury, Marcel G Alamán-Zárate, Danai Stefanidi, Emmanuel Maes, Hannah Fisher, Tomasz K Prajsnar and Stéphane Mesnage. Enterococcal cell wall remodelling underpins pathogenesis via the release of the Enteroccocal Polysaccharide Antigen (EPA). This image ca be can publish under the Creative Commons Attribution License (https://creativecommons.org/licenses/by/4.0/).
(JPG)

## Acknowledgments

The authors would like to thank Michel Arthur for the JH2–2 triple *pbp* mutant, Adelina Acosta-Martin for her help with LC-MS/MS experiments and Michelle Rowe for her technical support with NMR experiments. HR-MAS experiments were carried out at the NMR facility of the Advanced Characterization Platform of the Chevreul Institute (Villeneuve d'Ascq-France) by the staff of the PAGés platform.

## Author contributions

**Conceptualization:** Robert E Smith, Bartosz J Michno, Finn O'Dea, Jessica L Davis, Ian D.E.A. Lidbury, Tomasz K Prajsnar, Stéphane Mesnage.

**Data curation:** Robert E Smith, Bartosz J Michno, Rene L Christena, Finn O'Dea, Jessica L Davis, Emmanuel Maes, Hannah Fisher, Tomasz K Prajsnar, Stéphane Mesnage.

**Formal analysis:** Bartosz J Michno, Tomasz K Prajsnar, Stéphane Mesnage.

**Funding acquisition:** Robert E Smith, Rene L Christena, Finn O'Dea, Jessica L Davis, Tomasz K Prajsnar, Stéphane Mesnage.

**Investigation:** Robert E Smith, Bartosz J Michno, Rene L Christena, Finn O'Dea, Jessica L Davis, Ian D.E.A. Lidbury, Marcel G. Alamán-Zárate, Danai Stefanidi, Emmanuel Maes, Hannah Fisher, Tomasz K Prajsnar, Stéphane Mesnage.

**Methodology:** Robert E Smith, Bartosz J Michno, Rene L Christena, Finn O'Dea, Ian D.E.A. Lidbury, Marcel G. Alamán-Zárate, Emmanuel Maes, Hannah Fisher, Tomasz K Prajsnar, Stéphane Mesnage.

**Project administration:** Tomasz K Prajsnar, Stéphane Mesnage.

**Resources:** Bartosz J Michno, Tomasz K Prajsnar.

**Supervision:** Tomasz K Prajsnar, Stéphane Mesnage.

**Validation:** Robert E Smith, Bartosz J Michno, Finn O'Dea, Jessica L Davis, Tomasz K Prajsnar, Stéphane Mesnage.

**Visualization:** Robert E Smith, Bartosz J Michno, Rene L Christena, Finn O'Dea, Jessica L Davis, Ian D.E.A. Lidbury, Stéphane Mesnage.

**Writing – original draft:** Stéphane Mesnage.

**Writing – review & editing:** Robert E Smith, Bartosz J Michno, Rene L Christena, Finn O'Dea, Jessica L Davis, Ian D.E.A. Lidbury, Marcel G. Alamán-Zárate, Danai Stefanidi, Emmanuel Maes, Hannah Fisher, Tomasz K Prajsnar, Stéphane Mesnage.

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
