## [Decision Letter · Decision Letter 0]

PPATHOGENS-D-24-02546

Enterococcal cell wall remodelling underpins pathogenesis via the release of the Enteroccocal Polysaccharide Antigen (EPA)

PLOS Pathogens

Dear Dr. Mesnage,

Thank you for submitting your manuscript to PLOS Pathogens. After careful consideration, we feel that it has merit but does not fully meet PLOS Pathogens's publication criteria as it currently stands. Therefore, we invite you to submit a revised version of the manuscript that addresses the points raised during the review process.

Please submit your revised manuscript within 60 days Mar 20 2025 11:59PM. If you will need more time than this to complete your revisions, please reply to this message or contact the journal office at plospathogens@plos.org. Please include the following items when submitting your revised manuscript:

We look forward to receiving your revised manuscript.

Kind regards,

Christopher LaRock

Academic Editor

PLOS Pathogens

Helena Boshoff

Section Editor

PLOS Pathogens

 Sumita Bhaduri-McIntosh

Editor-in-Chief

PLOS Pathogens

orcid.org/0000-0003-2946-9497 Michael Malim

Editor-in-Chief

PLOS Pathogens

orcid.org/0000-0002-7699-2064

**Additional Editor Comments :**

Substantive concerns were raised over a number of critical issues. Several lines of new experiments are required before the paper could be reconsidered for publication. The opinion received from the reviewers and my own evaluation support my decision. Specifically, experiments 1) showing WT, the ΔatlE mutant and the complemented strain display differences in cell wall fragments, 2) more directly showing the presence of EPA or cell wall fragments in supernatant, 3) support for the decoy or another protective mechanism of action (eg resistance to antimicrobials and host innate immunity; like Smith et al., 2019 PLoS Path; Norwood et al., 2024, Mol Microbiol), and 4) showing sufficiency or cross-complementation of bacteria with purified cell wall fragments in the zebrafish model, are needed to support the core premise.

**Journal Requirements:**

At this stage, the following Authors/Authors require contributions: Robert E Smith, Rene L Christena, Bartosz J Michno, Jessica L Davis, Ian DEA Lidbury, Marcel Alaman-Zarate, Danai Stefanidi, Emmanuel Maes, Hannah Fisher, Tomasz K Prajsnar, and Stéphane Mesnage. Please ensure that the full contributions of each author are acknowledged in the "Add/Edit/Remove Authors" section of our submission form.

https://journals.plos.org/plospathogens/s/submission-guidelines#loc-parts-of-a-submission

- ® on page: 22 line 477.

5) In the online submission form, you indicated that "All data assocaited with this work are available upon request to the corresponding author. " All PLOS journals now require all data underlying the findings described in their manuscript to be freely available to other researchers, either

1. In a public repository

2. Within the manuscript itself

3. Uploaded as supplementary information.

Please note that your current contact point for data request is a co-author on this manuscript. According to our Data Policy, the contact point must not be an author on the manuscript and must be an institutional contact, ideally not an individual. Please revise your data statement to a non-author institutional point of contact, such as a data access or ethics committee, and send this to us via return email. Please also include contact information for the third party organization, and please include the full citation of where the data can be found.

7) Please amend your detailed Financial Disclosure statement. This is published with the article. It must therefore be completed in full sentences and contain the exact wording you wish to be published.

**Comments to the Authors:**

**Please note that one of the reviews is uploaded as an attachment.**

**Reviewers' Comments:**

Reviewer's Responses to Questions

**Part I - Summary**

Reviewer #1: In this study, the authors investigated the function of AtlE peptidoglycan hydrolase in Enterococcus faecalis. The experiments showed that AtlE acts together with AtlA to promote digestion of enterococcal chains at the stationary growth stage. The authors also demonstrated that AtlE is muramidase which is important for virulence in the zebrafish model of infection. The report is important in that the function of peptidoglycan hydrolases is generally under-investigated in enterococci.

However, the study did not prove that AtlE is directly involved in the release of cell wall fragments. It is also unclear if virulence in zebrafish models depends on EPA or peptidoglycan fragments. Related to the comments above, I don’t think the title is fully appropriate. Finally, there is no explanation if activity of AtlE in the release of cell wall fragments requires AtlA or other enzymes expressed in the stationary growth phase.

Reviewer #2: (No Response)

Reviewer #3: This is an interesting study by Smith et al that describes the discovery and activity of a new peptidoglycan hydrolase linked to surface polysaccharides in Enterococcus faecalis. This is relevant to E. faecalis pathogenesis as Epa is important for virulence and is linked to biofilm formation, antibiotic resistance, and phage resistance. In Ef, the peptidoglycan hydrolase AtlA is known to contribute to septum division and Ef chain length/morphology. A nice summary of AtlA activity is provided. The authors show that an atlABC knockout still forms diplococci/short chains in stationary phase, suggesting the presence of another peptidoglycan hydrolase active in this growth phase. The authors used zymography to identify peptidoglycan hydrolase activity from an atlABC deletion strain and performed mass spec to identify putative hydrolases. They confirm activity of a new hydrolase encoded in the variable region of the epa operon in E. faecalis and rename this gene product as AtlE. They show AtlE has a role in regulating chain length primarily in stationary phase, not exponential phase, although the effect is strongest when chain length is already misregulated in the absence of atlA (atlA atlE double mutant).

Interestingly, the authors show sequence and domain variability within the C-termini of AtlE variants from different strains and highlight strains OG1RF and JH2-2 as examples in this study (OG1RF encodes a DUF5776 domain whereas JH2-2 encodes a GW domain). The authors use recombinant AtlE from OG1RF to demonstrate that AtlE-OG1RF has N-acetylmuramidase activity on OG1RF cell walls, but not cell walls derived from JH2-2. The authors then show that AtlE-OG1RF has no activity against cell walls purified from a strain that produces modified Epa and posit that Epa decorations are required for AtlE activity. The authors then use NMR to show that AtlE does not affect surface display or structure of Epa in strain OG1RF. Finally, the authors demonstrate that AtlE affects virulence using an animal model. An atlE deletion model had attenuated virulence, and the authors propose that Epa-containing cell wall fragments released by AtlE may be important for virulence.

In general, this is a detailed manuscript highlighting a new peptidoglycan hydrolase that contributes to E. faecalis pathogenesis and is linked to the virulence factor Epa. The biochemical data supporting AtlE activity is clear. A minor weakness of the manuscript is the presentation of Figure 4, as the differences between AtlE-OG1RF and AtlE-JH2-2 are somewhat obscured. Given the differences in C-terminal domains between JH2-2 and OG1RF, and that AtlE-OG1RF did not have activity against cell wall fragments from JH2-2, it seems that the paper would benefit from an increased discussion of Epa and cell wall differences between these strains, and speculation of what the recognition motif is for AtlE.

**Part II – Major Issues: Key Experiments Required for Acceptance**

Reviewer #1: 1. Line 308. I disagree that the experiment in Fig 7 shows that the increase of embryo lethality is mediated by soluble cell wall fragments required EPA decoration. Firstly, this experiment should provide evidence that WT, the ΔatlE mutant and the complemented strain display differences in cell wall fragments. Secondly, it is unclear how WT or ΔepaR mutant cell wall fragments were obtained for this assay. Please, provide also details how the concentration was determined and why different concentrations of WT or ΔepaR were used. Thirdly, if cell wall fragments contain EPA attached to peptidoglycan, the effect of both components (EPA and peptidoglycan) should be studied in this virulence models. Lastly, the study with the soluble fragments should be conducted with all groups (OG1RF, WT, ΔepaR) in a single experiment. Thus, I believe that the statement in Abstract that soluble cell wall fragments containing EPA decorations increase the virulence of Enterococcus faecalis has not proved yet.

2. How deletion of AtlA and both enzymes, AtlA and AtlE, affect E. faecalis virulence in the zebrafish model? Do the enzymes work together to release cell wall fragments?

3. Could authors compare the activities of AtlA and AtlE in exponential and stationary growth stages?

4. Supplementary Fig 2 demonstrates that OG1RF supernatant is active with ΔponA ΔpbpF ΔpbpZ which was constructed in JH2-2 background. However, Fig 6B shows that AtlE is poorly active with JH2-2 cell wall material. Please explain this discrepancy.

Reviewer #2: (No Response)

Reviewer #3: 1) Based on data in Figure 7 and previously published work in reference 6, authors propose that cell wall fragments containing Epa are released into supernatants by AtlE, and that these fragments contribute to virulence. The authors show numerous zymograms demonstrating hydrolase activity in culture supernatants but it seems important to quantify or otherwise show the presence of Epa or Epa-containing cell wall fragments in supernatants.

**Part III – Minor Issues: Editorial and Data Presentation Modifications**

Reviewer #1: 1. Fig.1 A. Please indicate the growth conditions used to collect culture supernatants for this assay.

2. Fig. 2. Please show the whole gel for 36h conditions of renaturation. Indicate the growth conditions used to collect culture supernatants for this assay.

3. Please indicate the conditions for zymogram experiments in Supplementary Fig.1 and 2 (a renaturation time and the growth conditions used to collect culture supernatants). Please explain whether

4. Fig.3. FSC histograms of data depicted in Fig.3 should be provided in Supplementary material.

5. Fig. 3 A. Please indicate the concentration of ATc used to induce atlE expression.

6. Fig. 3 B. Please show the ΔatlA ΔatlE mutant.

7. Line 150. The authors conclude that the deletion of atlE alone was associated with a minor but significant increase in chain length in stationary phase. Please provide the analysis of the ΔatlE mutant complemented with atlE (statistics).

8. Line 306. Edit “he” typo.

9. Line 511. Zebrafish experiments. Please provide details of this experiment. The reference 20 is not correct. Indicate the growth conditions of bacterial strains used for this experiment. Describe how soluble cell wall fragments were obtained for zebrafish experiment and how the concentration of cell wall fragments was determined.

10. Reference Supplemental Fig 5,6, 7 in the text.

Reviewer #2: (No Response)

Reviewer #3: 1) Given that atlE is encoded in the epa operon, would an epa name not be more appropriate given the nomenclature used for other genes in this region?

2) It would be helpful to show a comparison of the entire Epa operon in OG1RF and JH2-2 with the location of atlE annotated. This would allow readers to put into context where atlE is with respect to other Epa genes.

2) Figure 1: Viewing or printing the manuscript in grayscale will obscure the red text. Using brackets and text directly in panel B may prevent confusion for readers.

3) Figure 3: A and B labels are missing from the figure panels.

4) Figure 4: Similar to the above comment, the color convention between panels A, B, and C may be lost to colorblind readers or if the manuscript is viewed in grayscale. Directly labeling OG1RF and JH2-2 in panel B with the DUF5776 and GW would be beneficial given that R1-R4 are used to label repeat region in both diagrams. In panel C, it is difficult to tell that there are 2 structures shown in the overlays. Colors with higher contrast (such as using black for OG1RF or JH2-2) would be helpful.

5) Lines 229-230: more info (1-2 sentences) on the biochemical nature of Epa decorations would be helpful here.

6) Figure S1: 0252 is listed twice in the legend (instead of 0114).

PLOS authors have the option to publish the peer review history of their article (what does this mean? ). If published, this will include your full peer review and any attached files.

**Do you want your identity to be public for this peer review?** For information about this choice, including consent withdrawal, please see our Privacy Policy .

Reviewer #1: No

Reviewer #2: No

Reviewer #3: No

**Figure resubmission:**
---

## [Decision Letter · Decision Letter 1]

PPATHOGENS-D-24-02546R1

Enterococcal cell wall remodelling underpins pathogenesis via the release of the Enteroccocal Polysaccharide Antigen (EPA)

PLOS Pathogens

Dear Dr. Mesnage,

Thank you for submitting your manuscript to PLOS Pathogens. After careful consideration, we feel that it has merit but does not fully meet PLOS Pathogens's publication criteria as it currently stands. Therefore, we invite you to submit a revised version of the manuscript that addresses the points raised during the review process.

Please submit your revised manuscript within 30 days Jul 22 2025 11:59PM. If you will need more time than this to complete your revisions, please reply to this message or contact the journal office at plospathogens@plos.org. Please include the following items when submitting your revised manuscript:

We look forward to receiving your revised manuscript.

Kind regards,

Christopher LaRock

Academic Editor

PLOS Pathogens

Helena Boshoff

Section Editor

PLOS Pathogens

Sumita Bhaduri-McIntosh

Editor-in-Chief

PLOS Pathogens

orcid.org/0000-0003-2946-9497

Michael Malim

Editor-in-Chief

PLOS Pathogens

orcid.org/0000-0002-7699-2064

**Additional Editor Comments:**

Please pay particular attention the need for genetic complementation to support conclusions on Epa release, as noted by two reviewers, the need for disclosing the interpretation of Staphylococcus data noted by Reviewer 2, and the additional minor writing issues noted by all.

**Reviewers' Comments:**

Reviewer's Responses to Questions

**Part I - Summary**

Reviewer #1: In the revised version of the manuscript, Robert Smith et al. have provided additional data and clarifications that convincingly address the key concerns about the original submission, and have substantially improved the quality of the manuscript.

Reviewer #2: Overall this is a well written manuscript that characterizes a new peptidoglycan hydrolase named AtlE that is conserved in Enterococci. It shows that AtlE affects cell chaining during stationary phase when the major AtlA is inactive, and that it displays N-acetylmuramidase activity. Cell wall degradation by this enzyme is strain specific and dependent on EPA decorations. When tested in a zebrafish systemic infection model, deletion of atlE leads to a decrease in lethality, and exogenous EPA-decorated cell wall fragments increase WT lethality.

Reviewer #3: This revised manuscript by Smith et al. describes identification of AtlE, an N-acetylmuramidase active in stationary phase in Enterococcus faecalis. The atlE gene is encoded in the biosynthetic locus for the enterococcal polysaccharide antigen Epa. The authors use a combination of genetics, biochemistry, NMR, and a zebrafish model to show the effect of AtlE on peptidoglycan, release of Epa into supernatants, and on E. faecalis virulence. Overall, this is a nice study that contributes knowledge about

The authors have responded to the reviewer comments and revised the manuscript appropriately. I still have a slight concern with the naming convention of atlE given the location in the genome but accept the author's rationale for keeping the atlE nomenclature. I also am unclear about the rationale for not providing the atlE complementation strain in Figure 8 given that several other figures include complementation. However, I think the other data supports the activity and biological importance of AtlE.

**Part II – Major Issues: Key Experiments Required for Acceptance**

Reviewer #1: (No Response)

Reviewer #2: The authors indirectly addressed most our previous comments, as many were quite similar to those of reviewer 1. However, it seems the authors did not recieve/see our comments (which we uploaded as an attachemetnt) since there is no point by point response to them.

Previous major comments listed here:

Major

1. The authors show that AtlE is important for control of cell chaining in stationary phase in the absence of AtlA. Can we consider AtlA and AtlE functionally redundant during stationary phase, where AtlE functions as a secondary autolysin that kicks in when AtlA is inactive? Is the LC-MS digestion profile of AtlE similar to cell wall digestions with AtlA? Drawing from what is known from studies with AtlB and AtlC, is there a possible physiologic explanation that would explain why the cell might require two autolysins during stationary phase?

2. The authors hypothesize that AtlE releases EPA to the environment and this in turn increases lethality of E. faecalis. While it is clear that AtlE and EPA-decorated cell wall fragments play a prominent role during systemic infection in the zebrafish model, the link between the two factors during in vivo infection has not been directly demonstrated. Have you tested if co-injection of WT EPA-decorated cell wall fragments restore lethality of the ∆atlE strain? And conversely, if AtlA and AtlE happen to be functionally redundant, do EPA-decorated cell wall fragments of the ∆atlE strain (released by AtlA) boost virulence of the WT strain?

3. At multiple points in the manuscript, the authors claim that the released cell wall fragments act as decoy molecules to inhibit the immune response, however no data are provided to substantiate this interpretation. Experiments to investigate should be performed. For example, previous work in this group has advanced our knowledge on the role of EPA during infection and interaction with macrophages. Do the EPA-decorated cell wall fragments inhibit the host immune system as opposed to the cell wall-attached EPA polymer. Do released EPA fragments protect from phagocytosis and charged antimicrobial peptides? This might help explain why the WT strain that already produces EPA-decorated cell wall fragments is more virulent when exogenous fragments are co-injected. Is it a matter of concentration?

Reviewer #3: A key tenet of the study is that AtlE affects virulence by modulating release of Epa. While the study now shows an effect on Epa release in the atlE mutant, this is not complemented. Several other figures present complementation data for atlE, so I am unclear about the rationale provided about why complementation was not feasible for showing Epa release. If it is not feasible to do the NMR experiments, even a relatively straightforward native gel stained with Alcian blue or Stains-All with polysaccharides precipitated from the supernatant would conclusively show complementation and strengthen the role of AtlE.

**Part III – Minor Issues: Editorial and Data Presentation Modifications**

Reviewer #1: 1. Fig. 9 B and C. Scale bar is missing.

2. Line 715. Indicate the method used to quantify cell wall fragments.

Reviewer #2: 1. In the response to reviewer 1, in the summary section, the comment about "a previous study showed that soluble peptidoglycan fragments have no impact on bacterial virulence in the zebrafish model of infection" is a misleading statement because the referenced paper refers to S. aureus, not E. faecalis.

2. In the response to reviewer 1, Mjaro issue comment 1, the authors inability to complement on a plasmid due to tetracylcine interferance can and should be overcome with chromosomal complementation, which is somewhat standard practice in the E. faecalis literature.

3. We note a number of typos in the discussion, so a thorough editing would be useful.

Previous minor comments listed here:

Minor

1. Since AtlE requires EPA decorations to hydrolyse peptidoglycan and AtlE is active during stationary phase, do you know if EPA surface exposure or EPA decorations are more abundant during stationary phase than during exponential phase?

2. Why was it necessary to use the cell wall substrate of a weakened cell wall strain for the zymogram assay in Fig. 1? Similarly, the OG1RF AtlE was able to digest the weakened cell wall of the JH2-2 strain in Figure S2, even though AtlE was later shown to be strain specific. Is this cross-reactivity due to the weak cell wall of this PBP mutant strain?

3. Fig. 3: why t test instead of ANOVA?

4. L139: typo “altA and altE”

5. L149: typo “led had”

6. L306: typo “he injection”

Reviewer #3: n/a

PLOS authors have the option to publish the peer review history of their article (what does this mean? ). If published, this will include your full peer review and any attached files.

**Do you want your identity to be public for this peer review?** For information about this choice, including consent withdrawal, please see our Privacy Policy .

Reviewer #1: No

Reviewer #2: No

Reviewer #3: No

**Figure resubmission:**
---

## [Editor Report · Decision Letter 2]

Dear Dr. Mesnage,

We are pleased to inform you that your manuscript 'Enterococcal cell wall remodelling underpins pathogenesis via the release of the Enteroccocal Polysaccharide Antigen (EPA)' has been provisionally accepted for publication in PLOS Pathogens.

Best regards,

Christopher LaRock

Academic Editor

PLOS Pathogens

Helena Boshoff

Section Editor

PLOS Pathogens

Sumita Bhaduri-McIntosh

Editor-in-Chief

PLOS Pathogens

orcid.org/0000-0003-2946-9497

Michael Malim

Editor-in-Chief

PLOS Pathogens

orcid.org/0000-0002-7699-2064
---

## [Editor Report · Acceptance letter]

Dear Dr. Mesnage,

We are delighted to inform you that your manuscript, "Enterococcal cell wall remodelling underpins pathogenesis via the release of the Enteroccocal Polysaccharide Antigen (EPA)," has been formally accepted for publication in PLOS Pathogens.

Best regards,

Sumita Bhaduri-McIntosh

Editor-in-Chief

PLOS Pathogens

orcid.org/0000-0003-2946-9497

Michael Malim

Editor-in-Chief

PLOS Pathogens

orcid.org/0000-0002-7699-2064